# MOTIONSTREAM: REAL-TIME VIDEO GENERATION WITH INTERACTIVE MOTION CONTROLS

**Joonghyuk Shin**[1,2] **Zhengqi Li**[2] **Richard Zhang**[2] **Jun-Yan Zhu**[3]

**Jaesik Park**[1] **Eli Shechtman**[2] **Xun Huang**[2,4]

[1]Seoul National University, [2]Adobe Research, [3]Carnegie Mellon University, [4]Morpheus AI

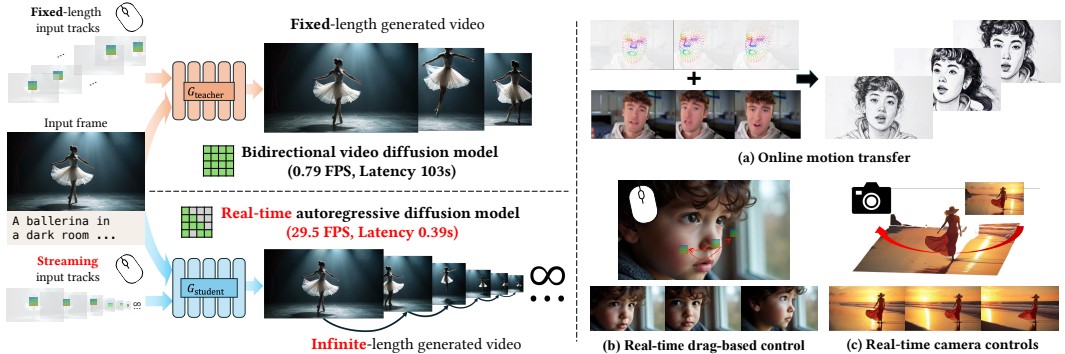

Figure 1: Prior motion-controlled video diffusion models typically operate offline to generate fixed-length sequences in parallel (top left). In contrast, our `MotionStream` enables streaming long-video generation from a single image with track control at interactive speed (bottom left). `MotionStream` can be applied to a variety of online downstream applications, such as real-time motion transfer, user drag operations, and 3D camera control (right).

## ABSTRACT

Current motion-conditioned video generation methods suffer from prohibitive latency (minutes per video) and non-causal processing that prevents real-time interaction. We present `MotionStream`, enabling sub-second latency with up to 29 FPS streaming generation on a single GPU. Our approach begins by augmenting a text-to-video model with motion control, which generates high-quality videos that adhere to the global text prompt and local motion guidance, but does not perform inference on the fly. As such, we distill this bidirectional teacher into a causal student through Self Forcing with Distribution Matching Distillation, enabling real-time streaming inference. Several key challenges arise when generating videos of long, potentially infinite time-horizons – (1) bridging the domain gap from training on finite length and extrapolating to infinite horizons, (2) sustaining high quality by preventing error accumulation, and (3) maintaining fast inference, without incurring growth in computational cost due to increasing context windows. A key to our approach is introducing carefully designed sliding-window causal attention, combined with attention sinks. By incorporating self-rollout with attention sinks and KV cache rolling during training, we properly simulate inference-time extrapolations with a fixed context window, enabling constant-speed generation of arbitrarily long videos. Our models achieve state-of-the-art results in motion following and video quality while being two orders of magnitude faster, uniquely enabling infinite-length streaming. With `MotionStream`, users can paint trajectories, control cameras, or transfer motion, and see results unfold in real-time, delivering a truly interactive experience.

# 1 INTRODUCTION

The ultimate goal of motion-controlled video synthesis is to give creators the power of a director's chair, allowing them to intuitively guide digital actors, objects, and cameras in real time. Although recent video diffusion models have made impressive strides toward this goal (Wang et al., 2024b; Geng et al., 2025; Burgert et al., 2025; Zhang et al., 2025b; Li et al., 2025b;c; Niu et al., 2024; Shi et al., 2024a; Zhou et al., 2025a; Lei et al., 2025), generating high-fidelity videos following user-specified motion trajectories, the experience remains far from interactive.

The promise of interactive control is currently hindered by several fundamental constraints. First, generation is too slow for interaction. For example, synthesizing a 5-second video clip with Motion Prompting (Geng et al., 2025) takes 12 minutes, trapping users in frustrating "render-and-wait" cycles. Second, the process is inherently non-causal, since diffusion models process the entire sequence in parallel with bidirectional attention. A user cannot see any partial results until the entire motion specification is complete. Finally, the inability to generate more than a few seconds of video severely limits the scope for any meaningful or extended creative expression. Together, these constraints (slow, non-causal, and short-duration generation) undermine the potential for a truly interactive creative experience.

To overcome these challenges, we introduce `MotionStream`, a method designed specifically for an interactive creative experience. Unlike conventional diffusion models that operate on the entire video sequence in parallel, `MotionStream` is an autoregressive model that synthesizes video in a streaming manner, reacting to user-drawn motion trajectories on-the-fly.

Our approach starts with a motion-controlled teacher model that uses lightweight sinusoidal embeddings with channel-wise concatenation for trajectory conditioning, avoiding the computational overhead of ControlNet-style (Zhang et al., 2023) architectures. Trained on both text and motion conditions, we introduce joint text-motion guidance that balances precise trajectory adherence with natural secondary motions enabled by text prompts. We then distill this teacher into a causal student through Self Forcing-style self-rollout (Huang et al., 2025b). While effective for short sequences, standard approaches drift during extended generation. Our analysis of attention patterns reveals persistent focus on initial frames alongside local temporal dependencies, similar to StreamingLLM (Xiao et al., 2023). This insight drives our attention sinking mechanism with rolling KV caches, which we incorporate directly into training to properly simulate inference-time extrapolation distributions, ensuring stable, indefinite-length generation at constant latency through fixed context windows.

`MotionStream` achieves 17 FPS at 480P and 10 FPS at 720P resolutions with sub-second latency on a single H100 GPU, reaching 29 FPS when optimized with efficient VAE decoders that we specifically train for streaming applications. Through extensive experiments and ablations, we demonstrate state-of-the-art performance across diverse motion control tasks including camera control, where our approach outperforms recent 3D methods while being more than $20\times$ faster. `MotionStream` transforms video generation from a passive waiting experience into an active creative process, where users can continuously interact with and guide the generation in real-time.

Our key contributions are:

1. We present the first *streaming* motion-conditioned video generation pipeline capable of running at 29.5 FPS on a single H100 GPU, enabling real-time interactive applications.

2. We propose a synergistic system harmonizing efficient architectural designs, including a lightweight track head and conditioning modules, with a distillation process that integrates joint text-motion guidance into the training objective, further accelerated by a Tiny VAE.

3. We introduce a distillation strategy for long video generation that systematically explores attention sinks and local attention with extrapolation-aware training for the first time, effectively preventing drift during long-term streaming.

4. Our approach achieves state-of-the-art results on motion transfer and camera control at orders of magnitude faster speeds, robustly generalizing to diverse interactive use cases.

## 2 RELATED WORK

**Controllable Video Generation.** Enabling precise user control is essential for applying video generative models to diverse downstream applications (Li et al., 2025d; Tu et al., 2025; Bahmani et al., 2024a; Gao et al., 2024c; Wu et al., 2025b; Fu et al., 2025). To this end, a large body of recent research has explored various types of control signals for video generation, such as structure control (Xing et al., 2024; Yang et al., 2025a; Jiang et al., 2025; Pang et al., 2024; Xing et al., 2025), camera control (Gao et al., 2024b; Zheng et al., 2024; He et al., 2024; Bai et al., 2025; Wu et al., 2025a; Yu et al., 2025b; Bahmani et al., 2024b; Yang et al., 2024b; Zheng et al., 2024; 2025), subject control (Huang et al., 2025a; Liu et al., 2025b; Fei et al., 2025; Liu et al., 2025a), and audio control (Tian et al., 2024; Gao et al., 2025; Peng et al., 2024).

As a unique modality that captures underlying video dynamics, motion has become a key conditioning signal for recent video diffusion models. Recent video diffusion models often condition generated videos on diverse forms of motion representations, including optical flow, 2D/3D motion trajectories, bounding boxes, and semantic segmentation (Goldman et al., 2008; Niu et al., 2024; Li et al., 2024; Wu et al., 2024b; Geng et al., 2025; Zhang et al., 2025b; Gillman et al., 2025; Shi et al., 2024a; Wu et al., 2024a; Gu et al., 2025b; Burgert et al., 2025; Tanveer et al., 2024). Despite their impressive quality, these methods are fundamentally limited to offline processing because they rely on diffusion models with full bidirectional attention, which requires the entire control signal to be known in advance. This constraint prevents their use in real-time, interactive applications.

**Autoregressive Video Models.** Early work adopted generative adversarial networks (GANs) for autoregressive or parallel video synthesis (Vondrick et al., 2016; Brooks et al., 2022; Villegas et al., 2017; Denton et al., 2017; Tulyakov et al., 2018; Liu et al., 2021; Li et al., 2022). More recently, there has been a paradigm shift towards using diffusion models trained with denoising objectives (Ho et al., 2022; Blattmann et al., 2023b; Yang et al., 2025b; Kong et al., 2024; Polyak et al., 2024; Blattmann et al., 2023a; Villegas et al., 2023; Deng et al., 2025; Gupta et al., 2024; Team Wan et al., 2025), or autoregressive (AR) models trained with next-token prediction (Weissenborn et al., 2020; Kondratyuk et al., 2024; Yan et al., 2021; Wang et al., 2024a; Bruce et al., 2024; Ren et al., 2025).

Another line of research integrates AR and diffusion to enable causal, high-quality video generation (Ruhe et al., 2024; Kim et al., 2024; Xie et al., 2024; Zhang & Agrawala, 2025; Sun et al., 2025; Weng et al., 2024; Liu et al., 2024; Chen et al., 2024; Guo et al., 2025b; Hu et al., 2024; Jin et al., 2025; Gu et al., 2025a; Gao et al., 2024a; Li et al., 2025f; Zhang et al., 2025a). Our work is inspired by the recent paradigm that distills a slow teacher model into a fast AR student for real-time performance (Yin et al., 2025; Huang et al., 2025b; Lin et al., 2025). However, these approaches either exhibit severe color drifts beyond the training horizon or require complex long-video finetuning, which poses challenges for controllable video generation.

**Interactive Video World Model.** Our work also belongs to interactive video world models, which aim to simulate environments for real-time interaction. This area has recently gained significant attention, as several recent works have shown impressive real-time, user-driven interaction (Ball et al., 2025; Li et al., 2025a; Yan Team, 2025; He et al., 2025; Bar et al., 2025; Po et al., 2025). However, most existing approaches either require substantial compute for inference (Ball et al., 2025; Parker-Holder et al., 2024), or are limited to closed-domain or synthetic environments (Yu et al., 2025a; Guo et al., 2025a; Yang et al., 2024a). In contrast, our work demonstrates that real-time, interactive generation for open-domain, photorealistic videos can be achieved on a single GPU.

## 3 MOTIONSTREAM: STREAMING GENERATION MEETS MOTION CONTROLS

Existing motion-conditioned video generation methods achieve strong motion-video alignment, but cannot support streaming interaction since bidirectional attention requires all future control signals upfront. Our proposed MotionStream addresses this through carefully designed causal distillation techniques, as illustrated in Figure 2. We first describe how to equip a pretrained video diffusion model with motion-control capability (Sec. 3.1) to serve as our bidirectional teacher, utilizing a lightweight track head and control modules designed to minimize architectural overhead. We then introduce our causal distillation pipeline, which performs extrapolation-aware training with attention sinks and local windows for long video generation, while integrating expensive joint text-

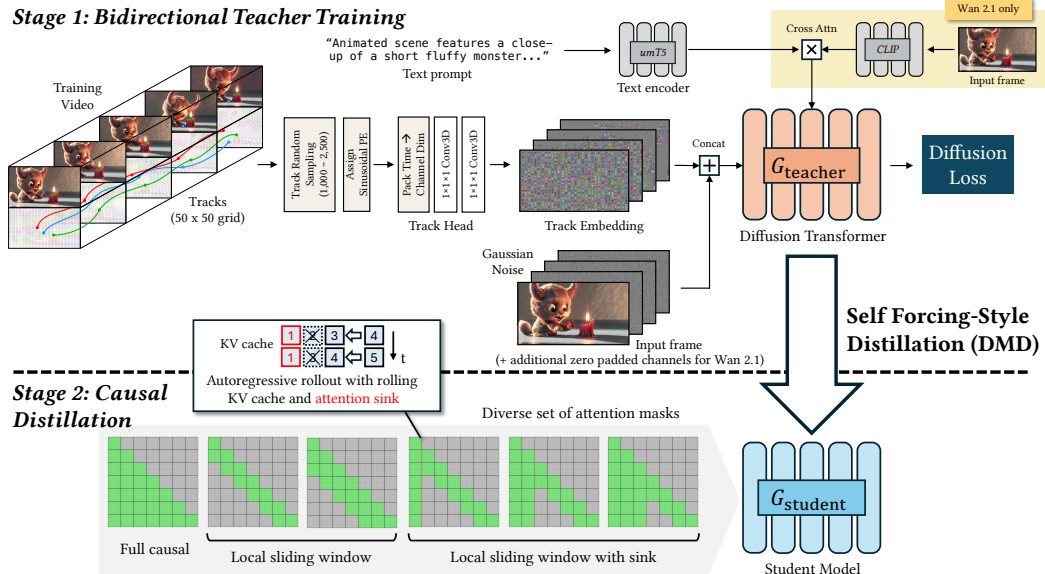

Figure 2: **Model architecture and training pipeline.** To build a teacher motion-controlled video model, we extract and randomly sample 2D tracks from the input video and encode them using a lightweight track head. The resulting track embeddings are combined with the input image, noisy video latents, and text embeddings as input to the diffusion transformer with bidirectional attention, which is then trained with a flow matching loss (top). We then distill a few-step causal diffusion model from the teacher through Self Forcing-style DMD distillation, integrating joint text-motion guidance into the objective, where autoregressive rollout with rolling KV cache and attention sink is applied during both training and inference (bottom).

motion guidance directly into the distillation objective for efficiency. Combined with our Tiny VAE, these joint efforts enable a highly responsive streaming experience.

## 3.1 ADDING MOTION CONTROLS TO BIDIRECTIONAL TEACHER MODELS

Training a high-quality motion-conditioned teacher model is important, as it determines both the quality upper bound and the architectural efficiency of the final distilled system. The teacher must also support diverse motion modalities, from complex real-world object dynamics to camera motions and user drags, which we achieve through the following design. We build our motion-guided teacher model on top of the Wan DiT family (Team Wan et al., 2025).

**Track Representation and Track Head Design.** Following MotionPrompting (Geng et al., 2025), each 2D track is assigned a unique $d$-dimensional embedding vector $\phi_n$, derived from a randomly sampled ID number through sinusoidal positional encoding. While encoding tracks as an RGB video and processing it through the CausalVAE is possible, we find that representing them with sinusoidal embeddings with a learnable track head achieves superior track adherence, video quality, and faster speed. We validate this in our experiments in Table 3. Given $N$ tracks $\{(x_t^n, y_t^n)\}_{n=1}^N$ across $T$ temporal frames, the input track-conditioning signal $c_m \in \mathbb{R}^{T \times H/s \times W/s \times d}$ is constructed by placing visible track embeddings at spatially downsampled locations, where $s$ is the VAE spatial downsampling rate and $v[t, n] \in \{0, 1\}$ indicates track visibility:

$$c_m \left[ t, \lfloor \tfrac{y_t^n}{s} \rfloor, \lfloor \tfrac{x_t^n}{s} \rfloor \right] = v[t, n] \cdot \phi_n. \tag{1}$$

Our lightweight track-encoding head performs $4\times$ temporal compression followed by a $1 \times 1 \times 1$ convolution. Prior methods adopt a ControlNet-style architecture (Zhang et al., 2023; Geng et al., 2025), which doubles FLOPs by duplicating network blocks. Instead, we directly concatenate the processed track embeddings with video latents, requiring only minor channel adjustments in the patchifying layer while leaving the core DiT architecture unchanged.

**Training.** We train the motion-guided teacher model through rectified flow matching objective (Liu et al., 2022; Lipman et al., 2022), where the forward process linearly interpolates between data $z_0$ and Gaussian noise $z_1 \sim \mathcal{N}(0, I)$: $z_t = (1-t)z_0 + tz_1$, $t \in [0, 1]$. The model is trained to predict the expected velocity fields with conditional flow matching loss $\mathcal{L}_{FM}$. One important limitation to note is that the model cannot inherently distinguish between occluded (non-visible) tracks and unspecified tracks, as both are represented by zero values. When a user releases controls during interaction, the model cannot determine whether the sudden zero values indicate occlusion or simply the absence of specification. This ambiguity occasionally leads to artifacts where objects abruptly appear or disappear. To address this issue, we introduce stochastic mid-frame masking with probability $p_{mask} = 0.2$, where $c_m[t_{rand}, :, :] = 0$, for randomly selected mid-frame chunks $t_{rand}$. In practice, we first train the model without masking to establish strong track-following capability, and then fine-tune with stochastic masking to preserve coherence when track signals change intermittently.

**Joint Guidance with Text and Motion Conditions.** Classifier-free guidance is an effective technique for steering diffusion models. We use both text and motion guidance and observe that they are complementary to each other. Text guidance generates natural dynamics but fails to maintain trajectory adherence. In contrast, track guidance enforces strict trajectory alignment but can produce overly simplistic and rigid motions, such as pure 2D planar translations in real-time drag scenarios. Therefore, we introduce a joint combination for simultaneous text and motion guidance:

$$\hat{v} = v_{base} + w_t \cdot \big(v(c_t, c_m) - v(\varnothing, c_m)\big) + w_m \cdot \big(v(c_t, c_m) - v(c_t, \varnothing)\big), \qquad (2)$$

where $v_{base} = \alpha \cdot v(\varnothing, c_m) + (1 - \alpha) \cdot v(c_t, \varnothing)$ and $\alpha = w_t / (w_t + w_m)$ (we omit $z_t$ for brevity). We find that the joint guidance weights ($w_t = 3.0$, $w_m = 1.5$) provide a good balance: text conditioning enables realistic dynamics, even with sparse, flat-grid inputs, while track guidance preserves trajectories and maintains shape fidelity. We further analyze these effects in Sec. 4.3. Although this increases sampling cost from 2 to 3 function evaluations (NFE) per denoising step in the teacher model, our causal distillation (described in the next section) embeds all guidance into a single NFE, thereby eliminating this overhead in the student model.

## 3.2 CAUSAL DISTILLATION

Existing motion-controlled video diffusion models require around 50 denoising steps to generate high-quality video and also presume that the entire input motion trajectory is known before generation. In this section, we distill the slow teacher model into a causal video diffusion model, enabling real-time streaming of long videos with motion control. Our training pipeline starts from the Self Forcing paradigm (Huang et al., 2025b), but off-the-shelf, we find this exhibits large latency fluctuations due to varying attention window sizes and performs well only within the teacher's training horizon (*i.e.*, 81 frames), with quality quickly degrading when extrapolating to longer sequences. This section presents several key technical innovations to address these issues.

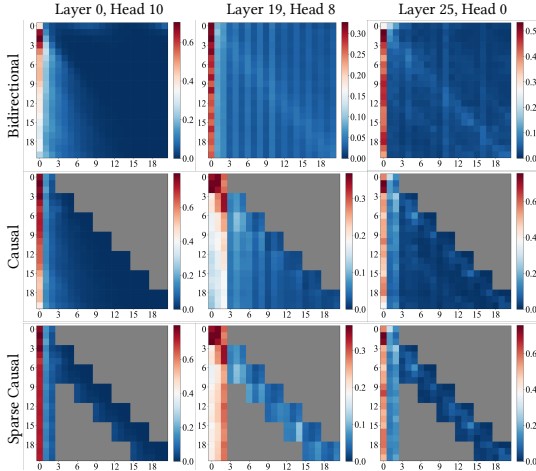

Figure 3: **Visualization of self attention probability map.** We visualize attention probability maps for bidirectional, full causal, and causal sliding window attentions. Several attention heads focus on the tokens corresponding to the initial frame throughout denoising generation.

**Attention sink in video diffusion model.** Autoregressive generation with sliding-window attention, as employed in Self Forcing (Huang et al., 2025b), is prone to quality degradation and drift during long-video extrapolation. To understand this failure, we visualize the self-attention maps in Figure 3 for both bidirectional and causal attention. Notably, many attention heads focus on the initial tokens corresponding to the input image. This phenomenon mirrors the observations in large language models Xiao et al. (2023), where initial tokens play a crucial role in stable streaming generation. Inspired by this, we adapt the attention sink concept to our video model by maintaining a local attention window while preserving the initial frame's tokens as a fixed anchor

during both training and inference. In our chunk-wise autoregressive setup, this sinking mechanism prevents the model from drifting during long-video extrapolation, even when trained only on short sequences limited by the teacher model.

**Causal Adaptation.** Following the initialization protocol from CausVid (Yin et al., 2025), we start our student model with the weights of the motion-guided teacher diffusion model and adapt it for causal attention architecture and few-step trajectory, using regression on ODE solution pairs sampled from the teacher. We train the model using attention masks with varying context window size and attention sink size for better generalization.

**Self Forcing-Style Distillation.** Following Self Forcing (Huang et al., 2025b), we perform temporal autoregressive roll-out with distribution matching distillation (DMD) (Yin et al., 2024b;a). During roll-out, video chunks are generated sequentially, conditioned on previously self-generated outputs rather than ground truth using KV cache.

We first define a video latent divided into $L$ chunks as $\{z_t^i\}_{i=1}^L$, where $t$ is the timestep. The sampling process of the $i$-th chunk can attend to its own noisy tokens and previously generated clean key and value tokens: $\mathcal{C}_i = \{z_t^i\} \cup \{z_0^j\}_{j \leq S} \cup \{z_0^j\}_{\max(1, i-W) \leq j < i}$, where $S$ denotes the number of sink chunks, and $W$ the attention window. Setting it to $\{z_t^i\} \cup \{z_0^j\}_{j < i}$ would correspond to a causal case with full history context window, matching that of Self Forcing. The generator $G_\theta$ produces each chunk autoregressively through the factorization $p_\theta(z_0^{1:L}) = \prod_{i=1}^L p_\theta(z_0^i | \mathcal{C}_i)$. Sampling each $z_0^i$ involves $K$-step iterative denoising on its latent $z_t^i$, starting from pure noise $z_T^i \sim \mathcal{N}(0, I)$.

In our approach, because the KV cache is continuously updated, RoPE values are assigned based on cache position rather than absolute temporal index. We apply KV cache rolling with attention sinks during both training and inference, implementing local attention without explicit attention masking and fully bridging the train–test gap, even in *extrapolation* scenarios. While TalkingMachines (Low & Wang, 2025) also explores attention sinks, our method differs in two key aspects that enhance long-video stability. First, we eliminate the train-inference mismatch by explicitly simulating the extrapolation process during training, using self-rollout with a rolling KV cache and attention sinks. In contrast, TalkingMachines employs synchronized denoising with causal attention masks, which does not fully replicate the dynamics of autoregressive inference. Second, our training process ensures the teacher model always evaluates continuous video frames. The setup in TalkingMachines introduces a temporal discontinuity between the sink frame and subsequent frames, which can push the input outside the teacher's pre-training distribution. By maintaining continuity, our teacher provides more robust scores for distillation.

After generating all $L$ chunks through self-rollout, we obtain the complete video $\hat{z}_0 = \{z_0^1, \ldots, z_0^L\}$. We then apply the DMD objective to this entire sequence, which minimizes the reverse KL divergence between the generator's output distribution and the data distribution: $\mathcal{L}_{\text{DMD}} = \mathbb{E}_t \left[ D_{\text{KL}}(p_t^{\text{gen}} \| p_t^{\text{data}}) \right]$. The gradient with respect to the generator parameters $\theta$ becomes:

$$\nabla_\theta \mathcal{L}_{\text{DMD}} \approx -\mathbb{E}_{t, \hat{z}_0} \left[ (s_{\text{real}}(\Psi(\hat{z}_0, t), t) - s_{\text{fake}}(\Psi(\hat{z}_0, t), t)) \cdot \frac{\partial \hat{z}_0}{\partial \theta} \right], \tag{3}$$

where $s_{\text{real}}$ is the score function for real data (approximated by the frozen bidirectional teacher) and $s_{\text{fake}}$ is the score function trained on the generator's outputs.

Intuitively, the driving gradient for the few-step causal generator $G_\theta$ comes from the difference between the estimated real and fake scores. To transfer the high-fidelity control of our joint guidance into the student without inference overhead, we define the target real score $s_{\text{real}}$ using the frozen teacher $f_\phi$ with joint guidance (omitting $z_t$ for brevity):

$$s_{\text{real}} = s_{\text{base}} + w_t \cdot (f_\phi(c_t, c_m) - f_\phi(\emptyset, c_m)) + w_m \cdot (f_\phi(c_t, c_m) - f_\phi(c_t, \emptyset)), \tag{4}$$

where $s_{\text{base}}$ follows the weighting defined in Eq. 2. In contrast, we parameterize the fake score $s_{\text{fake}}$ without using any CFG through a trainable critic $f_\psi$ (which approximates the generator's score): $s_{\text{fake}} = f_\psi(c_t, c_m)$. This configuration effectively "bakes" the computational cost of the teacher's multi-term guidance into the distillation objective, allowing the student generator to replicate the high-quality joint-guided distribution with a single function evaluation.

We update the generator ($G_\theta$) and fake score estimator ($f_\psi$) with a ratio of 1:5 (Yin et al., 2024a), allowing the critic to better approximate generated distributions. To reduce the memory usage during

Table 1: **Benchmark on Motion Transfer (Reconstruction).**

| Method | Backbone & Resolution | FPS | DAVIS Validation Set | | | | Sora Demo Subset | | | |
|---|---|---|---|---|---|---|---|---|---|---|
| | | | PSNR | SSIM | LPIPS | EPE | PSNR | SSIM | LPIPS | EPE |
| Image Conductor (Li et al., 2025e) | AnimateDiff (256P) | 2.98 | 11.30 | 0.214 | 0.664 | 91.64 | 10.29 | 0.192 | 0.644 | 31.22 |
| Go-With-The-Flow (Burgert et al., 2025) | CogVideoX-5B (480P) | 0.60 | 15.62 | 0.392 | 0.490 | 41.99 | 14.59 | 0.410 | 0.425 | 10.27 |
| Diffusion-As-Shader (Gu et al., 2025b) | CogVideoX-5B (480P) | 0.29 | 15.80 | 0.372 | 0.483 | 40.23 | 14.51 | 0.382 | 0.437 | 18.76 |
| ATI (Wang et al., 2025a) | Wan 2.1-14B (480P) | 0.23 | 15.33 | 0.374 | 0.473 | 17.41 | 16.04 | 0.502 | 0.366 | 6.12 |
| Ours Teacher (Joint CFG) | Wan 2.1-1.3B (480P) | 0.79 | **16.61** | **0.477** | **0.427** | **5.35** | **17.82** | **0.586** | 0.333 | **2.71** |
| Ours Causal (Distilled) | Wan 2.1-1.3B (480P) | **16.7** | 16.20 | 0.447 | 0.443 | 7.80 | 16.67 | 0.531 | 0.360 | 4.21 |
| Ours Teacher (Joint CFG) | Wan 2.2-5B (720P) | 0.74 | 16.10 | 0.466 | **0.427** | 7.86 | 17.18 | 0.571 | **0.331** | 3.16 |
| Ours Causal (Distilled) | Wan 2.2-5B (720P) | 10.4 | 16.30 | 0.456 | 0.438 | 11.18 | 16.62 | 0.545 | 0.343 | 4.30 |

self-rollout, we adopt Self Forcing's gradient truncation strategy: randomly sampling denoising step $k$ from $[1, K]$ and using the denoised output at k-th step as the final output, enabling gradient backpropagation only through that step. By additionally treating the cached KV from previous frames as stop-gradient context (detached), we track gradients only for the selected denoising step of the current block, effectively reducing the memory usage.

**Inference.** Our inference procedure identically follows the training process to ensure a perfect train-test match. We maintain a KV cache composed of $S$ chunks from the initial frames and a fixed-size local window of recent $W$ chunks. As new tokens are generated, this local window is "rolled" to maintain a constant size. For positional encoding, we store the final, RoPE-applied values for the static sink tokens, while tokens in the rolling window store pre-RoPE activations and receive positional indices dynamically based on their current cache location. Because this entire mechanism is simulated during training, the model seamlessly handles the discontinuity between sink and window tokens. This approach yields two key advantages over full-context methods: (1) initial image anchoring prevents drift during long rollouts, and (2) throughput and latency remain constant regardless of generated video length. We analyze the effects of attention sink and window size in Sec. 4.3 and describe our streaming pipeline in Sec. 4.4.

# 4 EXPERIMENTS

## 4.1 IMPLEMENTATION DETAILS

We build upon image-to-video (I2V) variants of Wan 2.1 (1.3B) and Wan 2.2 (5B) (Team Wan et al., 2025). We train teacher models on OpenVid-1M (Nan et al., 2024) and synthetic data generated by Wan text-to-video (T2V) model: Wan 2.1 at 832×480 (70K synthetic samples) and Wan 2.2 at 1280×704 (30K synthetic samples). For causal adaptation and Self Forcing distillation, we sample input images, text prompts and 2D tracks from the synthetic datasets described above. For computational efficiency, we track all real and synthetic videos from a 50×50 uniform grid with CoTracker3 (Karaev et al., 2024). We refer readers to Appendix for additional details.

## 4.2 QUANTITATIVE EVALUATIONS

**Motion Transfer.** We evaluate motion-following capability on two datasets: the DAVIS (Perazzi et al., 2016) validation set (30 videos) and 20 curated videos from the Sora webpage (Brooks et al., 2024). We use both because DAVIS presents challenging sequences with significant occlusions, while the Sora set provides clean examples with consistent visibility, ensuring a comprehensive evaluation. We directly compare the synthesized results with the corresponding ground-truth video frames. Visual fidelity is measured using PSNR, SSIM, and LPIPS (Zhang et al., 2018), while motion accuracy is assessed via End-Point Error (EPE), computed as the L2 distance between visible input tracks and the tracks extracted from the generated videos. All models are evaluated in their optimal configuration, and evaluations are performed at $832 \times 480$ resolution after resizing. Speed measures are based on a single H100 GPU. We refer readers to Appendix for detailed protocols.

**Camera Control.** To assess our model's ability to generate videos following camera controls, we evaluate its zero-shot performance on single-image 3D novel view synthesis. We compare it against several recent diffusion- and feed-forward-based view synthesis baselines (Zhou et al., 2025b; Yu et al., 2024; Xu et al., 2025) on the LLFF dataset (Mildenhall et al., 2019). To adapt our 2D track-controlled model for this task, we first es-

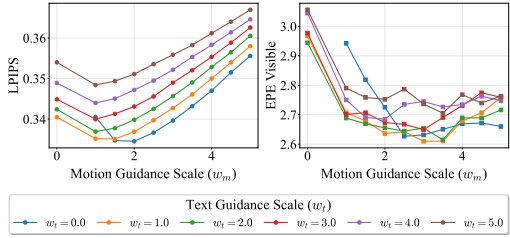

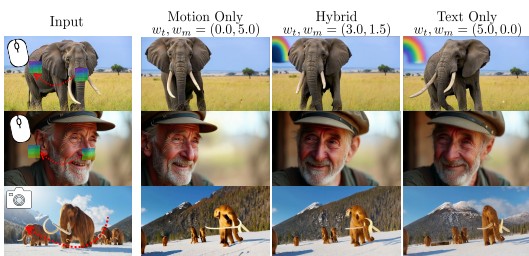

**Figure 4: Quantitative ablation on guidance.** We use Sora subset to ablate guidance strategies. Higher text guidance reduces overall metrics while motion guidance improves trajectory accuracy at the cost of visual quality (LPIPS).

**Figure 5: Qualitative ablation on guidance.** Pure motion guidance produces rigid movements while text guidance enables natural motion and shape preservation even with imperfect tracks. Our *Hybrid* joint guidance balances these two.

timate scene geometry using a monocular depth network (Wang et al., 2025b) and compute a single scale factor to align the predicted depth with the reconstructed scene point cloud from COLMAP (Schonberger & Frahm, 2016). We then derive 2D motion trajectories by interpolating between the input and target frame using depth and camera parameters.

For all methods, the input image aspect ratio is fixed at 16:9, and all synthesized frames are resized to $512 \times 288$ before evaluation with PSNR, SSIM, and LPIPS metrics. As shown in Table 2, our track-conditioned video generation models outperform other 3D novel view synthesis baselines by a large margin, even though they are not specifically designed for this task. Moreover, our causal models achieve significantly higher generation throughput compared with both the baselines and their bidirectional counterparts.

Table 2: **Evaluation on Novel View Synthesis.**

| Method | Resolution | FPS | LLFF | | |
|---|---|---|---|---|---|
| | | | PSNR | SSIM | LPIPS |
| DepthSplat (Xu et al., 2025) | 576P | 1.40 | 13.9 | 0.28 | 0.30 |
| ViewCrafter (Yu et al., 2024) | 576P | 0.26 | 14.0 | 0.30 | 0.30 |
| SEVA (Zhou et al., 2025b) | 576P | 0.20 | 14.1 | 0.30 | 0.29 |
| Ours Teacher (1.3B) | 480P | 0.79 | **16.0** | **0.42** | **0.21** |
| Ours Causal (1.3B) | 480P | **16.7** | _15.7_ | 0.38 | 0.23 |
| Ours Teacher (5B) | 720P | 0.74 | 14.0 | _0.40_ | _0.22_ |
| Ours Causal (5B) | 720P | _10.4_ | 15.0 | 0.39 | 0.23 |

## 4.3 ABLATION EXPERIMENTS

**Track Representation.** We compare our sinusoidal position encoding with a learnable track head against the RGB encoding strategy using a frozen VAE, following prior work (Gu et al., 2025b), where each 2D track is assigned a unique RGB color vector and placed onto a canvas before being fed into the VAE. Table 3 shows that our method (PE-Head) outperforms RGB-VAE in both efficiency and quality. Specifically, our lightweight PE-based encoding achieves better motion alignment while being two orders of magnitude faster than the VAE-based approach. We hypothesize sinusoidal encoding preserves stronger identification signals compared to RGB encoding due to richer expressive dimensions.

Table 3: **Comparing track representation methods.** Our sinusoidal PE with learnable track head outperforms RGB-VAE in both quality and efficiency, achieving 40× faster encoding critical for real-time streaming.

| Method | Time (ms) | DAVIS / Sora | | | |
|---|---|---|---|---|---|
| | | PSNR | SSIM | LPIPS | EPE |
| RGB-VAE | 1053 | 16.03 / 16.99 | 0.433 / 0.544 | 0.463 / 0.363 | 8.57 / 3.96 |
| PE-Head | **24.8** | **16.29 / 17.15** | **0.452 / 0.559** | **0.456 / 0.359** | **6.54 / 3.13** |

**Guidance Strategies.** Here we ablate our joint guidance approach from Sec.3.1. While pure motion guidance ($w_t = 0$, $w_m > 0$) achieves the highest trajectory accuracy, as shown in Figure 4, text guidance provides additional benefits for generating more diverse and realistic results. For example, text captions enable dynamics beyond trajectories alone, such as weather changes or object appearances as shown in the first row of Figure 5, which illustrates dragging an elephant while prompting "rainbow appears in background". Our empirical setting ($w_t = 3.0$, $w_m = 1.5$) balances motion fidelity with natural dynamics, adapting equally well to both precise and imperfect trajectories.

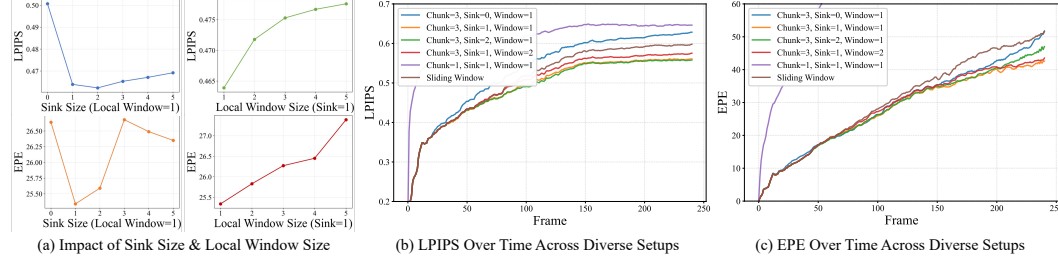

(a) Impact of Sink Size & Local Window Size  (b) LPIPS Over Time Across Diverse Setups  (c) EPE Over Time Across Diverse Setups

Figure 6: **Impact of Sparse Attention Patterns.** Using longer clips (up to 241 frames) from the Sora subset, we ablate attention sink size and local window size in extrapolation scenarios. Having at least a single sink chunk is crucial, but more provides marginal benefit, while larger window sizes degrade performance as attending to long-past history allows errors to accumulate in context tokens.

**Impact of Chunk Size, Attention Sink, and Window Size.** We investigate three key design choices that govern streaming quality and interactivity: the latent chunk size, the attention sink size, and the local context window size.

The chunk size, which defines how many video latents are processed in parallel, presents a critical trade-off between quality, latency, and throughput. As illustrated in the latency-throughput analysis in Figure A2, small chunks lead to lower throughput, while large chunks ($>3$) introduce prohibitive latency for real-time interaction. Additionally, as shown in Tab. 4, a chunk size of 1 causes significant quality degradation. We, therefore, select a chunk size of 3 as our optimal configuration, with further analysis provided in Appendix Sec. C.

To assess the impact of attention sinks and window size on model performance, we train a single model with randomly sampled sink and window sizes and then generate videos under different configuration combinations. We evaluate long-video extrapolation on Sora videos up to 241 frames (average 194 frames). Surprisingly, we find that the minimal configuration (a single-chunk sink with a single-chunk window)

Table 4: **Ablation study on Sora Extended.** c3s1w1 maintains high visual quality with small latency & throughput fluctuations, while vanilla sliding window exhibits large fluctuations degrading streaming stability.

| Config | Sora Extended | | | |
|---|---|---|---|---|
| | LPIPS | EPE | Latency (s) | Throughput (FPS) |
| Ours base (c3s1w1) | **0.464** | **25.34** | $0.70 \pm 0.01$ | $16.92 \pm 0.80$ |
| + Remove sink (c3s0w1) | 0.501 | 26.64 | $0.68 \pm 0.005$ | $\mathbf{17.43 \pm 0.88}$ |
| + Chunk Size 1 (c1s1w1) | 0.597 | 76.21 | $\mathbf{0.30 \pm 0.01}$ | $13.26 \pm 1.36$ |
| Sliding window | 0.480 | 28.09 | $0.80 \pm 0.08$ | $14.96 \pm 1.42$ |

achieves the best performance. Figure 6(a) shows that additional sinks provide only marginal gains while increasing latency, and expanded windows actually degrade performance. We hypothesize that this phenomenon arises because restricting the context to immediate predecessors, rather than long-past history, prevents error accumulation and thus reduces drift for long-video generation.

We additionally show reconstruction accuracy and throughput as well as their time evolution in Table 4 and Figure 6(b,c) respectively. We denote our configuration as chunk-3, sink-1, window-1 (abbreviated c3s1w1), while sliding window approach used in Self Forcing is c3s0w6, attending to a maximum of 6 previous chunks without sink tokens, with unbounded RoPE positions that scale with temporal frame count. Although removing sink tokens (c3s0w1) yields marginal speed improvements, this comes at the cost of degraded long-term generation stability.

## 4.4 Streaming Demo and Qualitative Results

For streaming demos, we further optimize the pipeline with an efficient Tiny VAE. Inspired by Boer Bohan (2025), we design and train a smaller VAE decoder with adversarial and LPIPS losses, regressing to the original VAE's latent space. Tiny VAE removes the VAE bottleneck by reducing the decoding time over $10\times$, improving Wan 2.1 from 16.7 FPS with 0.69s latency to 29.5 FPS with 0.39s latency, and Wan 2.2 from 10.4 FPS with 1.1s latency to 23.9 FPS with 0.49s latency on a single H100 GPU. We provide details for the tiny VAE in Appendix A, noting that the performance trade-off from switching to the tiny VAE is marginal in streaming scenarios, as shown

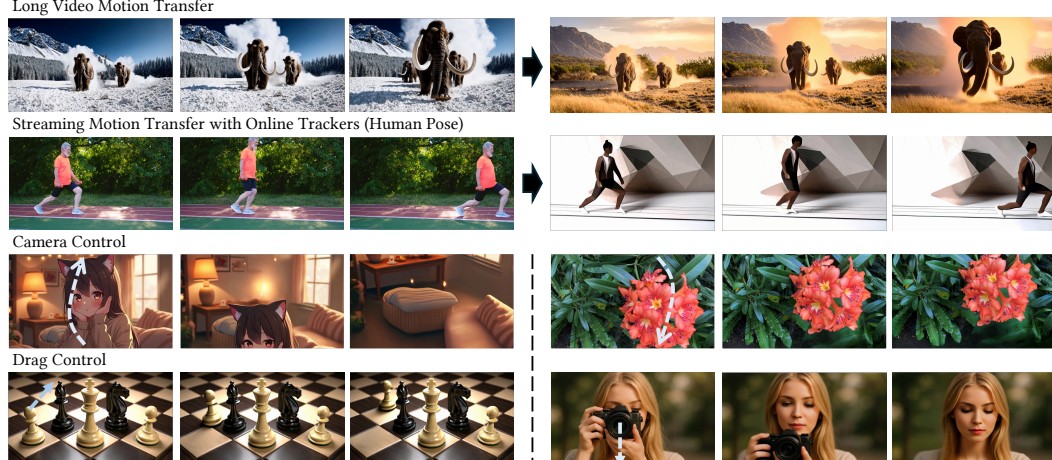

Figure 7: **Qualitative Results.** `MotionStream` can perform diverse downstream applications, including long video motion transfer (from offline or online sources), drag-based controls, and precise camera control with depth estimation. We showcase a few examples here.

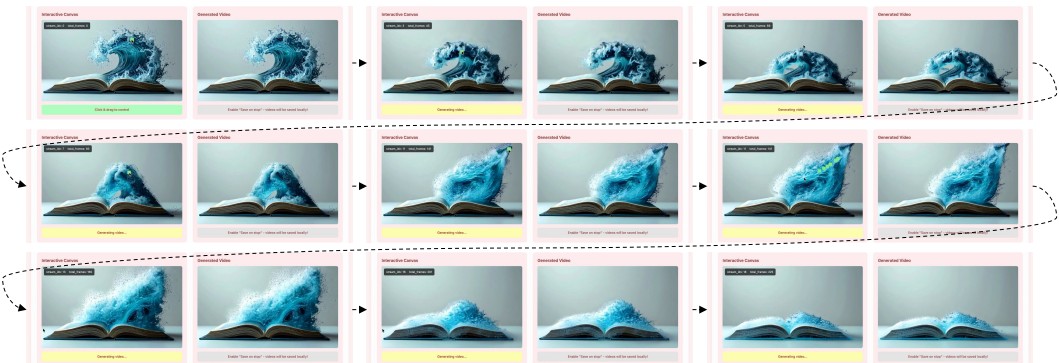

Figure 8: **Streaming Demo.** An example of real-time streaming generation; more details in Sec. C.

in Table A2. Leveraging track representation, our method enables causal, real-time execution of capabilities typically found in motion control approaches, such as mouse-based drag control and motion transfer using tracks from online trackers. Figure 7 demonstrates several of these applications. Additionally, we develop an interactive demo that enables real-time user control during streaming generation. Figure 8 shows an example, with further details in Sec. C and Figure A5.

## 5 CONCLUSION

We propose `MotionStream`, a framework for infinite-length video generation with interactive motion control, maintaining a stable 29 FPS on a single GPU. Our contributions span from training a motion-guided teacher with efficient track head and joint text-motion guidance, to distilling it into a causal student via self-rollout with attention sink and rolling KV caches. `MotionStream` achieves state-of-the-art results across diverse motion-conditioned generation tasks, while being significantly faster than prior methods. Limitations and future directions, as well as ethics and reproducibility statements are discussed in Appendix F and G.

**Acknowledgments.** This work was done at Adobe Research, during Joonghyuk Shin's internship and while Xun Huang was a Research Scientist. Jaesik Park was supported by the IITP grant No. RS-2021-II211343 (Artificial Intelligence Graduate School Program at Seoul National University, 5%), No. RS-2025-25442338 (AI Star Fellowship Support Program, 35%), and the NRF grant No. RS-2024-00405857 (60%) funded by the Korea government (MSIT). Jun-Yan Zhu was supported by the Packard Fellowship.

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

# A   TRAINING EFFICIENT TINY VAE

Table A1: **Comparison of Causal VAE Models.** We evaluate reconstruction quality on the Sora demo samples (resized to 81f×832×480) by encoding videos with the Full VAE encoder and decoding with different VAE variants. Our Tiny VAE achieves an order-of-magnitude faster decoding than Full VAEs while outperforming existing community implementations in reconstruction quality.

| Model | Decoder Params | Compression Rate | Decoding Time(s) (81×832×480) | PSNR | SSIM | LPIPS |
|---|---|---|---|---|---|---|
| Full VAE (Wan 2.1) | 73.3M | 8×8×4 | 1.67 | 31.43 | 0.934 | 0.069 |
| Tiny VAE (Wan 2.1  (Boer Bohan, 2025)) | 9.84M | 8×8×4 | 0.12 | 28.85 | 0.899 | 0.168 |
| Tiny VAE (Wan 2.1, Ours) | 9.84M | 8×8×4 | 0.12 | 29.27 | 0.904 | 0.107 |
| Full VAE (Wan 2.2, 5B) | 555M | 16×16×4 | 1.75 | 31.87 | 0.938 | 0.065 |
| Tiny VAE (Wan 2.2, Ours) | 56.7M | 16×16×4 | 0.23 | 28.43 | 0.883 | 0.126 |

As discussed in the main paper, the full CausalVAE becomes a bottleneck in streaming pipelines. Wan 2.1's CausalVAE performs $8\times$ spatial and $4\times$ temporal compression, while Wan 2.2 performs $16\times$ spatial and $4\times$ temporal compression. VAE decoding using the full CausalVAE takes 47% of chunk generation wall time for Wan 2.1 (1.3B), and 35% for Wan 2.2 (5B) in our base setup. Although Wan 2.2 (5B) VAE's higher compression rate enables Wan 2.2 to generate 720P videos in real time, it increases the decoding time and memory footprint.

Inspired by community implementations of Tiny VAE (Boer Bohan, 2025), we train a compact decoder from scratch with larger data, cleaner training pipelines, and better loss designs. While Boer Bohan (2025) trains Tiny VAE by regressing outputs to the original Wan VAE using adversarial loss from a PatchGAN discriminator and reconstruction loss with replay buffer, we extend this design by incorporating LPIPS loss with proper data scaling, and hyperparameter selections. We use a random subset of videos from OpenVid-1M and synthetic Wan videos (total 280K samples), training for 200K steps with a learning rate of $3 \times 10^{-4}$, batch size of 16, and AdamW optimizer. We train at a lower resolution of $144 \times 144$ and frame length of 21, but the model scales well to larger video dimensions.

As shown in Table A1, Tiny VAE achieves substantially faster decoding with significantly fewer parameters compared to the Full VAE. While Tiny VAEs typically produce slightly lower reconstruction quality than Full VAEs, our implementation substantially outperforms existing community versions. Importantly, when used jointly with our distilled student for latent decoding, we observe minimal quality differences in practice (Table A2), as most quality degradation and drift originate from the diffusion model itself rather than VAE reconstruction. For consistency when evaluating the performance of distilled diffusion model, we use results from Full VAE and report corresponding speed, and adopt Tiny VAE for the streaming demo.

Table A2: **Evaluating Tiny VAE in Streaming Generation Setup.** Using the same distilled student model, we ablate the impact of switching VAE from original Full VAE to Tiny VAE in Sora demo subset. It's important to note that even after changing to Tiny VAE, *our distilled models still outperform all other baselines* and quality differences compared to Full VAEs are marginal while achieving $1.75\times$ and $2.3\times$ higher throughput.

| Model | Throughput (FPS) | Latency (s) | PSNR | SSIM | LPIPS |
|---|---|---|---|---|---|
| Full VAE (Wan 2.1) | 16.7 | 0.69 | 16.67 | 0.531 | 0.360 |
| Tiny VAE (Wan 2.1, Ours) | 29.5 | 0.39 | 16.68 | 0.528 | 0.365 |
| Full VAE (Wan 2.2, 5B) | 10.4 | 1.14 | 16.62 | 0.545 | 0.343 |
| Tiny VAE (Wan 2.2, Ours) | 23.9 | 0.49 | 16.62 | 0.543 | 0.349 |

# B   VBENCH RESULTS AND USER STUDY

We additionally evaluate `MotionStream` using VBench-I2V (Huang et al., 2024) and conduct user studies on 20 samples from the Sora demo subset. In Vbench-I2V, we exclude camera motion

Table A3: **VBench-I2V Results.** We evaluate other baselines using VBench-I2V on Sora subset. While the results primarily depend on the choice of backbone, our models generally achieve high performance across all dimensions.

| Method | i2v subject | i2v background | subject consistency | background consistency | motion smoothness | aesthetic quality | imaging quality |
|---|---|---|---|---|---|---|---|
| Image Conductor (Li et al., 2025e) | 0.847 | 0.868 | 0.791 | 0.889 | 0.906 | 0.505 | 0.689 |
| GWTF (Burgert et al., 2025) | 0.957 | 0.974 | 0.933 | 0.944 | 0.981 | 0.620 | 0.675 |
| DAS (Gu et al., 2025b) | 0.972 | 0.987 | **0.953** | 0.958 | **0.988** | 0.634 | 0.695 |
| ATI (Wang et al., 2025a) | 0.981 | 0.988 | 0.948 | 0.947 | 0.980 | 0.629 | **0.707** |
| Ours Teacher (1.3B) | **0.984** | 0.988 | 0.948 | 0.943 | 0.987 | 0.625 | 0.698 |
| Ours Distilled (1.3B) | 0.982 | 0.987 | 0.940 | 0.941 | 0.985 | 0.618 | 0.684 |
| Ours Teacher (5B) | 0.983 | 0.988 | 0.947 | 0.959 | 0.982 | **0.637** | **0.707** |
| Ours Distilled (5B) | **0.984** | **0.990** | 0.945 | 0.959 | 0.987 | 0.630 | 0.703 |

and dynamic degree metrics since these dimensions are already constrained by the input trajectory conditions rather than text prompts. VBench results show strong correlation with the underlying backbone model, favoring recent Wan-based architectures. The provision of image and trajectory conditions leads to uniformly high scores across methods, reducing discriminative power between models. Nonetheless, both our teacher and distilled models consistently achieve competitive performance across all evaluated dimensions.

For user study, we collected 2,800 responses evaluating video quality of generated videos using our Wan 2.1 (1.3B) variants. Since accurately assessing track-following capability from thousands of grid point trajectories is challenging for participants, we focused solely on video quality assessment, with results shown in Figure A1. As with VBench, video quality correlates strongly with backbone capacity. Notably, ATI (Wang et al., 2025a), which leverages Wan 2.1 14B (10× larger than our 1.3B model), generally produces more visually favorable videos. However, despite ATI's aesthetic quality, we observe it often lacks precise trajectory adherence. Both our teacher and student models outperform other baselines in quality, with the teacher being slightly preferred over the student.

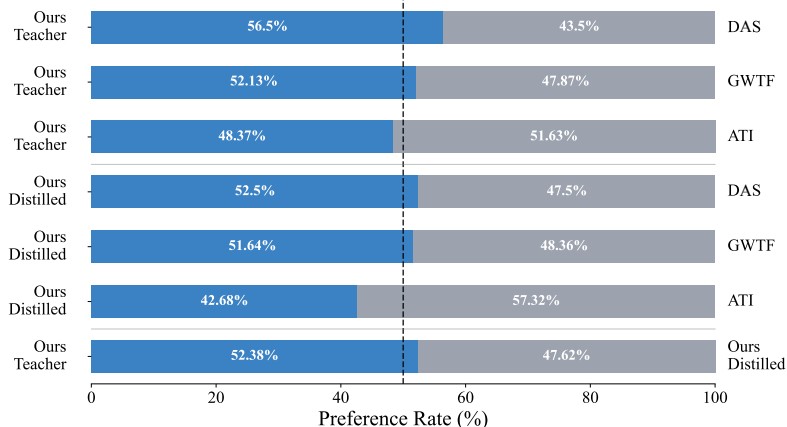

Figure A1: **User Study Results.** We evaluate video quality through pairwise comparisons on 20 Sora samples. In terms of pure video quality, our models outperform all baselines except ATI, which uses a 10× larger backbone (Wan 2.1-14B), producing visually favorable videos.

## C ADDITIONAL ABLATION EXPERIMENTS AND QUALITATIVE RESULTS

**Impact of Chunk Size and Sampling Steps.** In the main paper, we primarily focused on evaluating the efficiency of diverse sparse attention patterns with attention sink size and attention window size. As shown in Table 4 and Figure 6, reducing the chunk size increases the number of autoregressive rollouts required to generate the same length of video while reducing the amount of bidirectional attention, leading to worse performance. Since chunk sizes beyond 3 induce high latency, which makes them less useful for our interactive use case, we report only speed metrics for these configu-

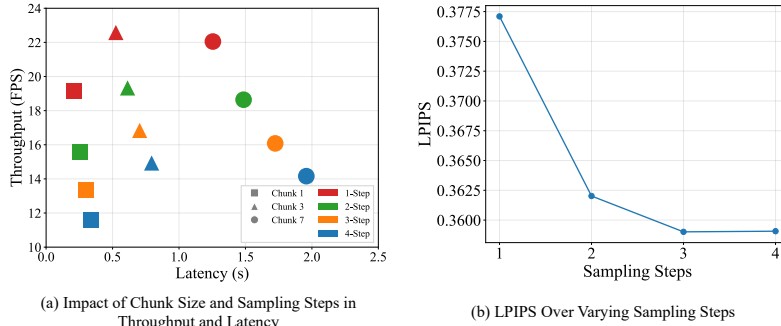

(a) Impact of Chunk Size and Sampling Steps in Throughput and Latency

(b) LPIPS Over Varying Sampling Steps

Figure A2: **Speed and Quality Tradeoffs with Chunk Sizes and Sampling Steps.** We visualize the latency-throughput relationship across varying chunk sizes and sampling steps (left), and image quality (LPIPS) across different sampling steps, using our default setup of `c3s1w1` (right).

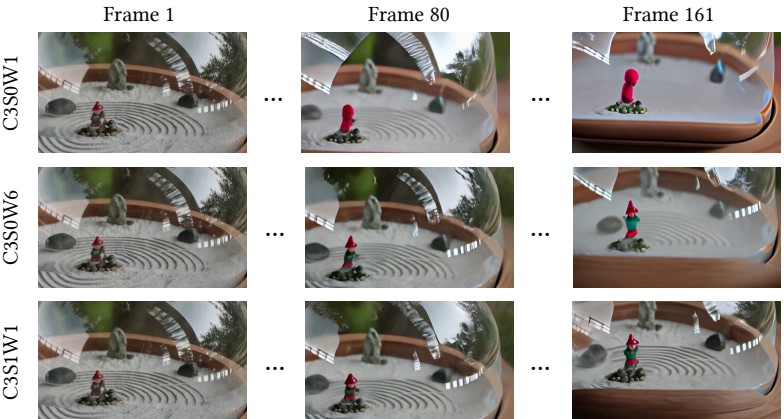

Figure A3: **Long video extrapolation with and without attention sink.** Models without attention sink (top two rows) exhibit cumulative drift over time, while our approach with attention sink (bottom) maintains stable quality throughout extrapolation.

rations. While we did not perform controlled training ablations for larger chunk sizes, we observe similar trends with TalkingMachines (Low & Wang, 2025), where quality generally improves with larger chunks at the cost of latency. Figure A2 (a) visualizes the latency and throughput given different chunk sizes and sampling steps. Note that since all benchmarks are conducted with a sink size and a window size of 1, the model with chunk size 7 has lower FPS than the one with chunk size 3 because its attention sequence length is longer.

The number of sampling steps per chunk also affects this balance. We train our student model for 3-step generation, which we find to be optimal for our motion-controlled setup; increasing beyond 3 steps yields only marginal quality gains, while reducing to 2 steps causes a noticeable drop in quality, as shown in Figure A2 (b). Although trained for 3-step inference, the DMD framework allows for flexible sampling configurations at test time. Based on these findings, we use a chunk size of 3 with 3 sampling steps as our primary configuration to achieve a strong balance between interactivity and high-quality video generation.

**Visualization of Attention Sink's Impact on Long Video Extrapolation.** We present long video extrapolation results comparing generation with and without attention sinks. As shown in Figure A3, incorporating at least one sink chunk proves crucial for preventing drift during extended generation. Without this, the model exhibits increasing degradation over time, while the attention sink enables stable quality maintenance throughout the video sequence. Please refer to the videos in the supplementary materials for additional results.

**Impact of Motion Control on Generative Capability.** To assess whether injecting motion control compromises the base model's generative capability, we compare our 1.3B models against the

Table A4: **Impact of Motion Control on Generative Quality.** We evaluate whether injecting motion control degrades the pretrained model's capability by comparing against a larger, dedicated I2V baseline (Wan 2.1 14B I2V). We also report performance when motion conditions are dropped. Results indicate that adding motion conditioning does not significantly degrade the base model's generative quality. While removing motion conditions introduces a slight quality drop as our models were not optimized for this setting, the output still adheres to the given text and image inputs.

| Model | i2v subject | i2v bg | subj. consis. | bg consis. | motion smooth. | aesth. qual. | imag. qual. | FVD ($\downarrow$) |
|---|---|---|---|---|---|---|---|---|
| Wan 2.1 I2V (14B) | 0.979 | 0.987 | 0.947 | **0.953** | 0.988 | 0.619 | **0.711** | 1274.6 |
| Ours Teacher (1.3B) w. Motion | **0.984** | **0.988** | **0.948** | 0.943 | 0.987 | **0.625** | 0.698 | **578.2** |
| Ours Teacher (1.3B) w/o Motion | 0.965 | 0.976 | 0.889 | 0.926 | **0.990** | 0.613 | 0.707 | 1550.4 |
| Ours Student (1.3B) w. Motion | 0.982 | 0.987 | 0.940 | 0.941 | 0.985 | 0.618 | 0.684 | 745.8 |
| Ours Student (1.3B) w/o Motion | 0.973 | 0.982 | 0.923 | 0.943 | 0.986 | 0.597 | 0.688 | 1532.5 |

larger Wan 2.1 14B I2V baseline, which serves as a high-quality upper bound (as no Wan 1.3B I2V exists) on the Sora dataset. As shown in Table A4, we observe no noticeable degradation; despite the capacity gap and open-source training data, performance remains robust. In fact, a few demo samples demonstrate that motion control enables the model to generate dynamic motions that naive I2V models often struggle to produce due to static bias (Choi et al., 2025). Consequently, FVD scores are lower with motion conditions, as the generated videos better mimic the target distribution through explicit motion constraints.

We also investigate the model's behavior when motion conditions are not provided. Since our models are incentivized to accurately follow motion and are not explicitly trained for an I2V (or "motion-less") setup, providing empty motion inputs indeed results in a quality degradation. Nonetheless, while the quality is slightly lower compared to the fully conditioned setting, the model still follows text prompts effectively without severe visual collapse. One interesting observation is that the teacher model without motion conditions rarely produces sudden scene changes, yet results in a lower subject consistency metric. We hypothesize that applying CFG with empty motion conditions leads to unstable outputs where text prompts dominate. We did not observe this behavior with guidance-distilled student models. Please refer to the supplementary videos for detailed results.

**Qualitative Comparison.** We also provide a qualitative comparison between baselines in Figure A4. Please refer to the supplementary videos for detailed results.

**Streaming Demo.** We show some examples of our streaming demo in Figure A5. The demo starts by accepting an input image and text prompts, which can help generate effects that are not achievable through mouse drags. Users can then choose the specific spacing/size of the track grids and start controlling objects in a scene, or move the camera. Due to its autoregressive nature, users can pause or resume the streaming generation process. Users can start/end or pause/resume the generation process with `Enter` and `Space` keyboard input, which is especially useful for dynamically adding static grids to specify unmoving regions or multiple moving grids to control different motions during streaming. As a video generative model, our method naturally supports drag-based image editing, generating intermediate transition frames as a bonus, while being faster than most dedicated drag-based image editing methods (Pan et al., 2023; Shi et al., 2024b; Nie et al., 2023; Shin et al., 2024; Zhao et al., 2024). To further support diverse downstream tasks, we will continuously update our front-end UI with additional features.

# D   TRAINING DETAILS

**Data Preprocessing.** We train our models on two primary data sources: OpenVid-1M (Nan et al., 2024) and synthetic videos generated by larger Wan text-to-video models. For OpenVid-1M, we filter the dataset to 0.6M videos by requiring a minimum of 81 frames and a 16:9 aspect ratio, sampling at 16 FPS. For synthetic data, we use 70K samples for Wan 2.1 (81 frames, 480P resolution, generated by Wan 2.1 14B using text prompts from VidProm (Wang & Yang, 2024)) and 30K publicly available samples for Wan 2.2 (121 frames, 720P resolution, generated by Wan 2.2 5B from the FastVideo team (The FastVideo Team)).

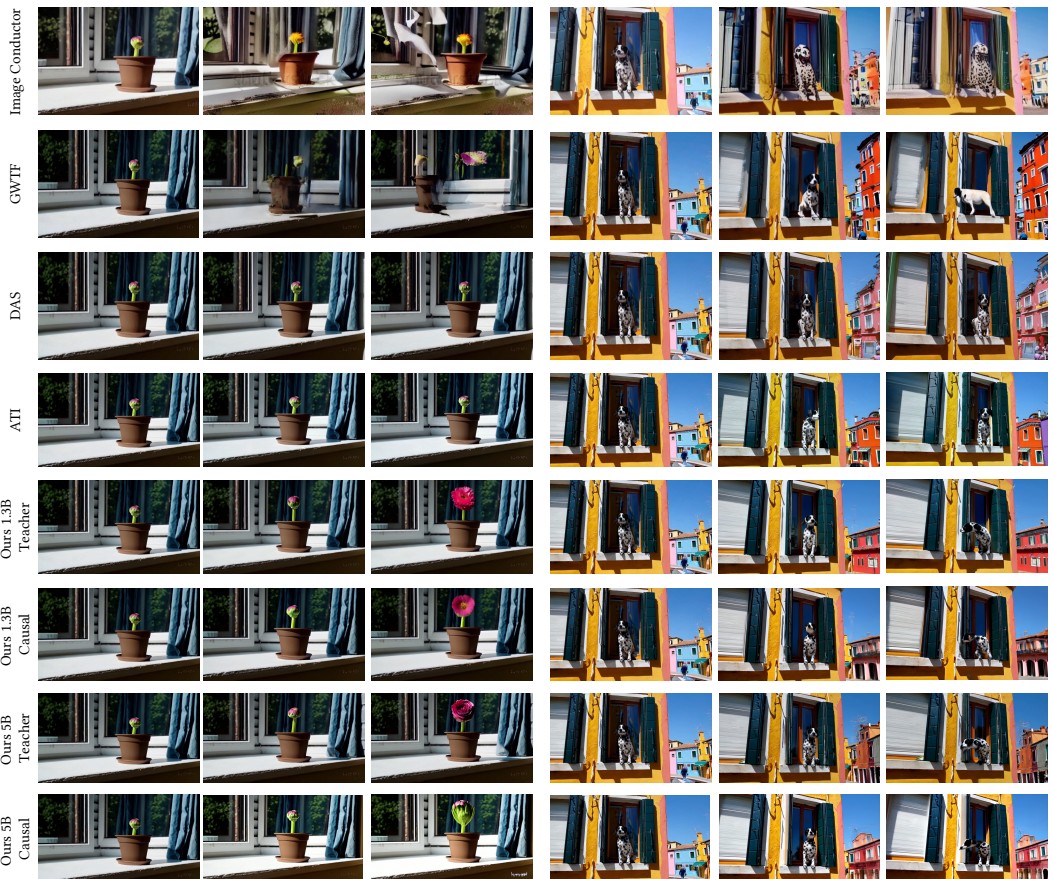

Figure A4: **Qualitative Comparison.** Generated videos from the Sora subset. As seen on the left, our model successfully reconstructs the flower blooming motion, while GWTF captures the motion but suffers quality degradation. ATI produces high-quality videos but with reduced trajectory adherence in motion transfer scenarios.

We extract motion trajectories from all videos using CoTracker3 (Karaev et al., 2024), tracking points on a 50×50 uniform grid. Our training follows a two-stage design: initial training on OpenVid-1M to establish general motion conditioning capability, followed by fine-tuning on cleaner synthetic data to improve trajectory adherence and reduce artifacts from noisy real-world videos. For the distillation phase, we use only synthetic samples since the distillation process converges quickly and requires less data.

For causal adaptation, we generate 4,000 training samples using our joint guidance strategy ($w_t = 3.0, w_m = 1.5$). Notably, Self Forcing-style distillation with DMD objectives does not require complete video sequences during training due to the nature of distribution matching loss design, requiring only the first frame, text prompt, and corresponding motion tracks.

**Teacher Model Training.** We initialize our teacher models from partial weights of VideoXFun's Wan variants (AIGC-Apps & Alibaba PAI Team, 2024), which extend Wan I2V models with additional control channels. This initialization accelerates convergence compared to training from scratch. Both Wan 2.1 and Wan 2.2 undergo two-stage training: (1) initial training on filtered OpenVid-1M (0.6M videos) for 4.8K steps, followed by (2) fine-tuning on cleaner synthetic data for 800 steps (Wan 2.1) and 400 steps (Wan 2.2), approximately one epoch each. During training, we randomly sample 1,000~2,500 tracks and assign a sinusoidal positional embedding of $d = 64$ dimensions. Stochastic track masking (described in Sec. 3.1) is applied during the fine-tuning stage. We use batch size 128 with learning rates of $1 \times 10^{-5}$ and $1 \times 10^{-6}$ for stages 1 and 2, respectively. The track head remains frozen after initial training as it already operates chunk-wise.

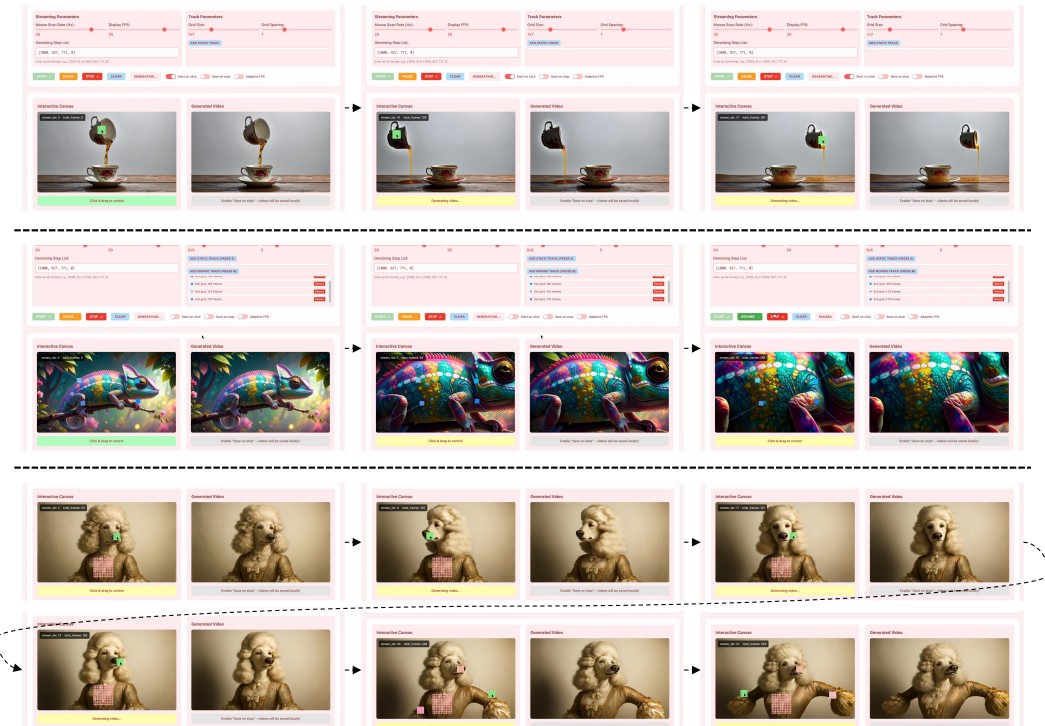

Figure A5: **Streaming Demo.** We show some examples of our streaming demo. Green grids indicate the points that are being dragged (online), red grids indicate a set of points that should remain static, and blue grids indicate user's pre-drawn trajectories for moving multiple points simultaneously (see chameleon example). More examples can be found on our supplementary website.

We train the motion-guided teacher model through rectified flow matching objective, where the forward process linearly interpolates between data $z_0$ and Gaussian noise $z_1 \sim \mathcal{N}(0, I)$:

$$z_t = (1 - t)z_0 + tz_1, \quad t \in [0, 1], \tag{A1}$$

with timestep shifting (Team Wan et al., 2025), defined as $t'(k, t) = \frac{kt/1000}{1+(k-1)(t/1000)}$, where $k = 6.0$. The model is trained to predict the expected velocity fields conditioned on noisy latents, denoising time steps, text prompts $c_t$ and motion embedding $c_m$, with timestep-dependent weighting $w_t$:

$$\mathcal{L}_{\text{FM}} = \mathbb{E}_{z_0, z_1, t} \left[ w_t \left\| v_\theta(z_{t'}, t', c_t, c_m) - (z_1 - z_0) \right\|^2 \right]. \tag{A2}$$

**Causal Architecture Adaptation.** Following CausVid (Yin et al., 2025), we adapt to few-step causal inference through ODE trajectory regression. We generate 4,000 ODE trajectories from the teacher model and train for 2,000 steps. While the transition from bidirectional to causal attention requires significant adaptation, we find variations within causal patterns (e.g., different window sizes) are similar enough to learn jointly with single model. We therefore train with diverse sparse causal attention masks, creating a unified initialization that supports flexible self-rollout configurations. We maintain a batch size of 128 with a learning rate $2 \times 10^{-6}$.

**Self Forcing-Style Distillation with DMD.** Self Forcing distillation converges quickly at around 400 steps with a batch size of 64. We set learning rates to $2 \times 10^{-6}$ for the generator and $4 \times 10^{-7}$ for the critic (fake score function), with a 1:5 update ratio and gradient truncation as described in Sec. 3.2.

We employ AdamW optimizer (Kingma & Ba, 2015; Loshchilov & Hutter, 2019) with mixed precision (bfloat16) and PyTorch's FSDP. Exponential moving average (EMA) is applied during teacher training and DMD distillation. Wan 2.1 variants train at $832 \times 480$ resolution while Wan 2.2 trains at $1280 \times 704$. With 32 A100 GPUs, training the Wan 2.1 teacher model takes roughly 3 days with

causal adaptation and distillation completing in 20 hours, while Wan 2.2 requires slightly longer training times.

# E    EVALUATION PROTOCOLS

Since different methods employ various backbone models with different spatial and temporal resolutions, we optimize the evaluation setup for each method by matching their primary spatial/temporal dimensions. For DAVIS evaluation, when the number of frames exceeds a model's default temporal length, we retain the first and last frames while uniformly subsampling intermediate frames, then compare against correspondingly subsampled ground truth frames.

All Sora demo videos were limited to 81 frames at 16 FPS for standard experiments, except for the extrapolation experiments in Table 4 and Figure 6, which use up to 241 frames (average 194 frames, approximately 15 seconds). Similar subsampling was applied for models with shorter temporal contexts (16 frames for AnimateDiff, 49 for CogVideoX).

We report the best scores for each method using its optimal configuration. For Image Conductor, we tested with 1, 10, 100, 1000, 2500 tracks and report results from 100 tracks, which performed best. Go-With-The-Flow requires dense optical flow as input, so we provide flow estimated by RAFT following their reference implementation. Diffusion-As-Shader uses 3D tracks as input, for which we provide tracks from SpatialTracker (Xiao et al., 2024) on a $70 \times 70$ grid (their default setting). ATI was tested with 40, 2500 tracks, and we report results from 40 tracks (their default), which performed better. Our models consistently use 2D trajectories from $50 \times 50$ initial grid points tracked by CoTracker3.

After generation, all results are resized to $832 \times 480$ resolution to ensure a consistent scale for metrics, particularly EPE which calculates L2 distance between track coordinates. All latency and throughput are measured using a single H100 GPU in bfloat16 precision with Flash Attention 3 (Shah et al., 2024).

For the camera control benchmark on the LLFF dataset, we derive $50 \times 50$ grid tracks using the method described in Sec. 4.2. To minimize unintended object motion and focus the evaluation purely on camera movement, we use the prompt template: "static scene, only camera motion, no object is moving, {scene_name}", where {scene_name} is replaced with the specific LLFF scene name.

# F    LIMITATION AND FUTURE WORK

While MotionStream achieves real-time motion control for long-range video generation, we identify several limitations. First, the fixed attention sink mechanism, while ensuring stable long-term generation, constrains the model's ability to handle scenarios with complete scene changes. Our approach maintains strong anchoring to the initial chunk, which works well for most motion-controlled generation scenarios where cameras and objects move within consistent environments. However, when presented with trajectories from game engines or other sources where environments change continuously, the model exhibits a tendency to preserve the initial scene rather than adapting to new contexts. This limitation is also inherent to current 2D tracking systems, which cannot meaningfully track and encode complete scene transitions. Future work could explore dynamic attention sinking strategies that adaptively refresh anchor frames for world modeling applications.

Second, we observe artifacts when motion trajectories are extremely rapid or physically implausible, manifesting as temporal inconsistencies or distortions in object appearance. One good approach for future work would be exploring effective track augmentation strategies during training to better simulate imperfect user inputs and scaling to larger backbone models, which generally exhibit more robust visual quality.

Lastly, our pipeline sometimes struggles to preserve source details when scenes, text prompts, or intended motions are highly complex. This primarily stems from backbone capacity limitations. While motion conditioning with text prompts can enforce movements beyond what the base model generates from text alone, quality may be unsatisfactory in such cases. We also note that different image conditioning mechanisms across different backbones can affect the robustness of the model in handling imperfect motion cues. Interestingly, we empirically observed that the smaller Wan 2.1 (1.3B)

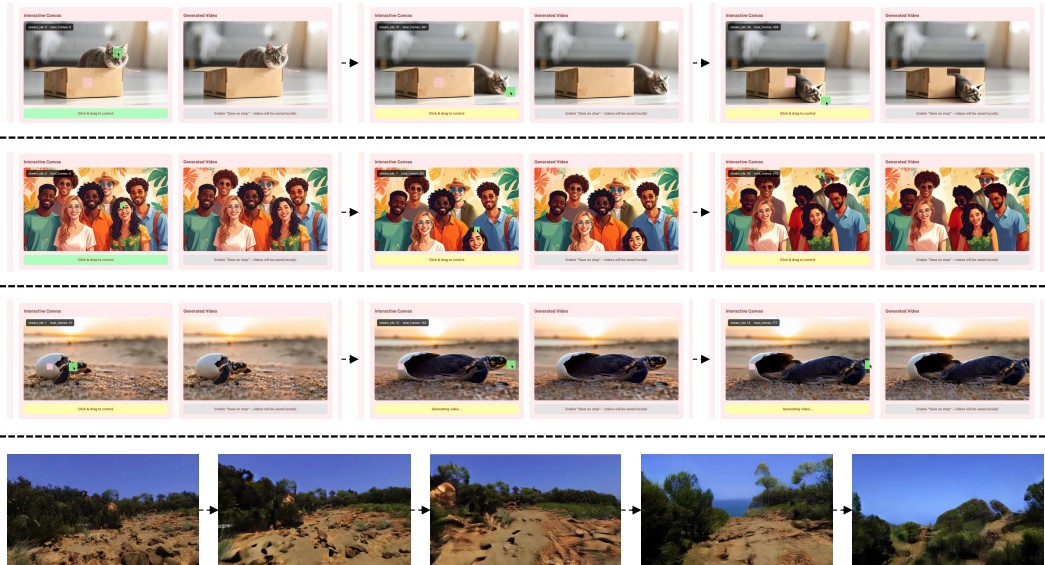

Figure A6: **Failure Cases.** In the cat and turtle examples, the intention was to bring the cat out of the box and make the turtle hatch from the egg. However, due to limitations in hand-drawn trajectories for expressing complex motions and the backbone model's generalization capacity, the outputs are not physically plausible, and objects are either deformed or simply translated along the tracks. In complex scenes with multiple human identities (second example), the model often loses identity information and produces artifacts. The last row shows our model's output on the world exploration task. Since track representation struggles to capture complete scene transitions and the attention sink prioritizes preserving source features, our pipeline faces difficulty in scenarios where new objects appear or scenes continuously change. More videos can be found on our supplementary website.

usually outperforms Wan 2.2 (5B) in preserving source structures, particularly with user-drawn flat-grid imperfect point trajectories. We attribute this to Wan 2.1's input image cross-attention design which helps maintain the original structure throughout, while Wan 2.2's TI2V structure is slightly more experimental. As our models are relatively modest in scale, we expect larger base models to provide improved performance and stability under challenging scenarios. Some of our failure cases can be found in Figure A6.

## G  ETHICS STATEMENT

As video generative models become increasingly capable of producing realistic content that mimics world dynamics, we recognize the potential for misuse. While `MotionStream` advances inter-active content creation with intuitive controls, our approach naturally inherits potential risks from the underlying generative technology, including the creation of deceptive media. We emphasize the critical need for parallel development of safeguarding techniques such as watermarking, content authentication, and controlled access mechanisms. We encourage prioritizing responsible deployment strategies alongside capability improvements to ensure these tools benefit society while minimizing harms.

**Reproducibility.** Our models are built upon publicly available Wan model variants and datasets (OpenVid-1M and synthetic Wan videos). Training was conducted in a well-controlled environment, with all training details, hyperparameters, and implementation specifics provided in Sec. D and the supplementary to ensure reproducibility.

**LLM Usage.** We used LLMs to help polish the writing and presentation of this manuscript. LLMs were not used for research ideation, experimental design, or scientific discovery in this work.

