# OpenReview forum: "MotionStream: Real-Time Video Generation with Interactive Motion Controls"
_ICLR.cc/2026/Conference — ICLR 2026 Oral_

### Official Review · Reviewer_RuEW · 2025-10-30

**Soundness:** 3
**Presentation:** 3
**Contribution:** 3
**Rating:** 6
**Confidence:** 3

**Summary:**

This paper presents MotionStream, a real-time motion-controlled video generation system that achieves sub-second latency at 24 FPS on a single GPU, representing a two-order-magnitude speedup over existing methods. The approach distills a bidirectional motion-conditioned text-to-video teacher model into a causal student model using Self Forcing with distribution matching loss, enabling streaming inference. To address the challenges of infinite-horizon generation, the system introduces sliding window causal attention combined with attention sinks and KV cache rolling during training, which allows it to generate arbitrarily long videos at constant speed while maintaining quality and preventing error accumulation. This enables truly interactive experiences where users can control videos through trajectory painting, camera movements, or motion transfer and see results unfold in real-time, achieving state-of-the-art results in both motion following and video quality.

**Strengths:**

1. This paper achieves sub-second latency for motion-controlled video generation at 24 FPS on a single GPU, representing a two-order-magnitude speedup over existing methods through causal distillation of a bidirectional teacher model using Self Forcing with distribution matching loss.

2. This paper enables infinite-length video generation at constant speed by introducing sliding window causal attention with attention sinks and KV cache rolling, which prevents quality degradation and error accumulation while maintaining fixed computational costs regardless of video length.

3. This paper delivers truly interactive motion control experiences by allowing users to paint trajectories, control cameras, or transfer motion with real-time visual feedback, achieving state-of-the-art results in both motion following and video quality.

**Weaknesses:**

1. The technical contributions are somewhat limited. The proposed MotionStream seems to be a straightforward combination of existing full-attention controllable video generation with the self forcing framework.
2. While the paper claims to enable long video generation, only a single long-form demonstration is provided; the authors should present more extensive examples to substantiate the robustness and consistency of their approach across diverse scenarios.

**Questions:**

1. The authors acknowledge that fixed attention sinks limit the model's ability to handle complete scene changes, favoring initial scene preservation; they should provide concrete visual examples demonstrating this failure case to better illustrate when and how the method breaks down under drastic environmental transitions.
2. In Figure 5, when conducting the guidance ablation study, the authors use different guidance scales for the same conditioning across experiments, such as w_m=1.5 in Hybrid versus w_m=5.0 in Motion Only; could the observed performance differences be primarily attributed to these inconsistent hyperparameter settings rather than the conditioning strategies themselves?

---

> ### Author Response · Authors · 2025-11-22
>
> Dear reviewer RuEW, we are grateful for the valuable comments. Please refer to our general response for a summary of contributions, the link to the [website](https://anonysubmission7ei3vc8j3x1f3d-spec.github.io/anony-submission-suppleweb/) containing additional samples, and the updated manuscript. Here, we address your specific concerns.
>
> **(1) Limited technical contribution**
>
> We kindly refer to our summary of contributions in the general response. As discussed, our method is not a straight forward combination of bidirectional video model and self forcing distillation. We addressed several critical challenges, including the efficient design of motion conditioning modules and mitigating the drifting issue during long video generation via extrapolation-aware rollouts with attention sinks and rolling kv cache. As a result, MotionStream enables highly practical downstream applications as illustrated on the website, which are distinct from typical world exploration or T2V models.
>
> **(2) Additional samples (long form, scene changes)**
>
> Our demo samples are mostly over 300 frames and several of them are well beyond 500 frames (e.g., Cute monster, Elephant, Hot airballoons, Man, Shiba dog, Paper boat, Woman, etc). We play the demo at 30 FPS, so it would be almost twice as long if we ran at 16 FPS which is Wan's default speed. We acknowledge a trade-off inherent to using fixed attention sinks and short attention windows: balancing source detail preservation with consistent physical memory. In world exploration tasks where conditioning tracks disappear rapidly, 2D track representation also faces inherent limitations, often causing the model to revert to initial scene layout when signals vanish. Please refer to the “Failure Cases” tab on our website for examples in the world exploration domain. Nonetheless, we note that these characteristics serve as advantages in our primary drag-based control setups (maintaining source details, consistent ~30 FPS, and accurate motion conditioning), which typically feature minimal drastic scene transitions. Our evaluation datasets (DAVIS/SORA) also contain several challenging scene changes; please refer to the response to Reviewer g9qk and the “Additional Evaluation Examples” tab on the webpage for more details.
>
> **(3) Different CFG strength for motion and track**
>
> It’s quite common to have distinct optimal guidance strengths for different conditioning modalities. For instance, InstructPix2Pix [1], which introduced nested CFG with image and text condition, adopted text guidance values between 5-10 and image guidance values between 1-1.5. We performed a grid search for both motion guidance and text guidance scales within the range of [0, 5] and provided all quantitative results as a graph in Figure 4. Then, based on quantitative metrics and empirical observation after distillation target (where high motion guidance resulted in overly simplistic translational movements), we set the text and motion guidance scale as 3.0 and 1.5 respectively for optimal experience.
>
> [1] Brooks et al., “InstructPi2xPix: Learning to Follow Image Editing Instructions”, CVPR 2023.

---

### Official Review · Reviewer_5Qbe · 2025-10-31

**Soundness:** 3
**Presentation:** 2
**Contribution:** 2
**Rating:** 6
**Confidence:** 3

**Summary:**

This paper presents MotionStream, a real-time, motion-controlled video generator that achieves sub-second latency and up to 24 FPS on a single GPU, enabling interactive dragging, motion transfer, and camera control with effectively unbounded video length. The model conditions on sparse 2D tracks using lightweight sinusoidal embeddings and a learnable track head, and employs joint classifier-free guidance to balance strict trajectory adherence with text-prompted natural dynamics. Long-horizon stability is maintained via sliding-window causal attention supplemented by attention sinks and rolling KV caches, which anchor early frames while keeping computation constant. The authors substantiate the approach with extensive comparisons, ablations, and a user study, demonstrating clear advantages for efficient motion-controlled video generation.

**Strengths:**

1. The manuscript’s motivation is clear and compelling, and the proposed solution appears technically sound and promising.
2. The authors provide thorough comparative evaluations and ablation studies that isolate the contributions of each component.
3. The presented demos are persuasive, showcasing the method’s practical impact.

**Weaknesses:**

1. The manuscript now reads like a composition of prior work—self-forcing, attention sinks, and motion conditioned video generation—assembled to achieve real-time, controllable video generation. However, specific contributions are not clearly presented.
2. For motion conditioning, the method employs channel-wise concatenated embeddings and fine-tunes pretrained image-to-video models. While computationally efficient, this design may degrade the pretrained model’s general generative capabilities. The manuscript lacks analysis or experiments that quantify the trade-offs introduced by this specialization.
3. The manuscript does not include comparisons of general image-to-video quality with the original pretrained backbone. Such an evaluation would provide a more complete picture of the method’s impact on baseline capabilities.
4. The evaluation omits challenging corner cases—such as abrupt, large-amplitude motion changes or trajectories that poorly match the source subject. Moreover, because the approach combats long-horizon self-forcing with attention sinks and a sliding-window scheme tied to the teacher’s training horizon (e.g., position embeddings capped at ~81 frames), the model may be constrained in realizing large or sudden motion transitions. The authors should explicitly test and discuss these scenarios to clarify whether the architectural choices impose limits on motion magnitude or adaptability.

**Questions:**

1. In the guidance ablation, why do the text-only generations still exhibit a clear motion trajectory in the visualizations? Is this because the training data contain many similar side-to-side sway patterns, or is there another cause?
2. In Table 1, models with more parameters perform worse than smaller ones. The manuscript does not explain this.

---

> ### Author Response · Authors · 2025-11-22
>
> Dear reviewer 5Qbe, thanks for your detailed comments and feedback. Please refer to the general response for a summary of contributions, the [website](https://anonysubmission7ei3vc8j3x1f3d-spec.github.io/anony-submission-suppleweb/) link for additional samples, and the updated manuscript. Here, we address your specific concerns.
>
> **(1) Clarification on the contributions**
>
> Please refer to our general response for a clarification of contributions. Beyond achieving real-time, interactive video generation, we also explore the efficient design of motion conditioning modules, and methods to mitigate the drifting issue during long video generation via extrapolation-aware rollouts with attention sink and rolling kv cache. We have revised the overall manuscript for clarity. Please let us know if you would like to have specific sections edited for better presentation.
>
> **(2) Preserving base model’s generative capability**
>
> We thank the reviewer for pointing this out. We provide additional experiment results to validate whether 1) injection of motion control degrades the pretrained model’s generative capability, and 2) whether MotionStream can still perform image-to-video (I2V) generation without motion conditions.
>
> Since there is no comparable I2V baseline for Wan 2.1 1.3B (we initialized with partial weights from Wan Fun [1]), we compare against the much larger Wan 2.1 14B I2V model to establish a high-quality upper bound. We evaluate using 20 representative Sora prompts and videos, reporting VBench I2V results and FVD.
>
> | Model | i2v_subject | i2v_bg. | subject_consis. | bg_consis. | motion_smooth. | aesthetic_qual. | imaging_qual. | fvd |
> |:---:|:---:|:---:|:---:|:---:|:---:|:---:|:---:|:---:|
> | Wan 2.1 I2V (14B) | 0.979 | 0.987 | 0.947 | **0.953** | 0.988 | 0.619 | **0.711** | 1274.6 |
> | Ours Teacher (1.3B) w. Motion | **0.984** | **0.988** | **0.948** | 0.943 | 0.987 | **0.625** | 0.698 | **578.2** |
> | Ours Teacher (1.3B) wo. Motion | 0.965 | 0.976 | 0.889 | 0.926 | **0.990** | 0.613 | 0.707 | 1550.4 |
> | Ours Student (1.3B) w. Motion | 0.982 | 0.987 | 0.940 | 0.941 | 0.985 | 0.618 | 0.684 | 745.8 |
> | Ours Student (1.3B) wo. Motion | 0.973 | 0.982 | 0.923 | 0.943 | 0.986 | 0.597 | 0.688 | 1532.5 |
>
> * **Does injection of motion control degrade the pretrained model?**
> We do not observe noticeable quality degradation from injecting motion control, as supported by the experiment results. Considering factors such as capacity difference between 14B and 1.3B, fine-tuning on relatively low quality open-source data, and inevitable drop from causal distillation, the performance remains robust. In fact, some qualitative demos show that motion control enables the model to generate dynamic motions which naive I2V models often struggle to due to static bias [2]. Consequently, FVD scores are noticeably better (lower) with motion conditions, as the generated videos better mimic the target distribution through explicit motion constraints.
>
>
> * **Does MotionStream work without any motion condition (I2V)?**
> Our training objective incentivizes accurate following of motion conditions, so we did not optimize specifically for pure I2V usage. As noted in Section 3.1, we randomly drop motion conditions during training to simulate intermittent user inputs (e.g., releasing the mouse), but not to the extent of a full unconditional I2V setting. Nonetheless, while quality is slightly lower compared to the fully conditioned setting, the model still follows text prompts effectively without visual collapse. One interesting observation is that the teacher model without the motion condition rarely produces sudden scene changes, resulting in a notably lower subject consistency metric. As our model is not trained for this scenario, we hypothesize that applying CFG with pure text condition with empty motion results in unstable outputs where the text prompt often dominates. We did not observe this behaviour with guidance-distilled student models. Please check the “I2V Results (No Motion Cond.)” tab of our websites for video samples. To fully preserve the base I2V model’s capability, one might also consider training LoRA, toggling it for strength; however, we did not explore this direction as we had to start from fine-tuning all modules, due to the lack of Wan 2.1 1.3B I2V model.

---

> ### Author Response · Authors · 2025-11-22
>
> **(3) Evaluation on challenging cases**
>
> We have added several demo samples featuring challenging motions (e.g., rapid mouse drags) and additional hard samples from the DAVIS and SORA datasets. Please refer to the "Additional Evaluation Examples" and "Failure Cases" tabs on our website. We respectfully note that both DAVIS and SORA datasets contain scenarios with rapid motion, significant perspective changes, and large deformations; our quantitative results confirm that our method still generally outperforms baselines in these cases. Please check the response to the reviewer g9qk for additional discussions. In general, given the characteristics of track condition and attention sink, our method is more suited to animating objects within a scene, with moderate camera view changes rather than full world exploration tasks.
>
> We understand the reviewer’s concern in potential limitation with long-horizon motion as the teacher only captures motion distribution capped at 81 frames. In our approach, KV cache rolling occurs continuously for videos longer than 3 latent chunks. Since this mechanism is explicitly simulated during training, the model learns to handle extrapolation naturally and we did not face noticeable constraints during longer inference. Our streaming demos contain drags lasting much longer than 81 frames, yet motions remain smooth and natural because every extrapolation step remains within the training domain.
>
> We also considered training with longer videos. However, point trajectories mostly become highly noisy and usually disappear after 100 frames, especially with noisy open-source videos, making conditioning tricky. Nonetheless, we believe performing distillation with larger backbones that support longer video sequences would allow simulation of more roll-outs during training and could provide potentially more informative gradient signals.
>
>
> **(4) Text-only guidance showing motion trajectory**
>
> In the guidance ablation, “text-only” means that classifier-free guidance (CFG) is applied only in the text domain. Referring to Equation (2) in the paper, setting $w_ m=0$ yields $\hat{v} = v(\varnothing, c_ m)+ w_ t \cdot \big( v(c_ t, c_ m) - v(\varnothing, c_ m) \big)$. Thus motion condition $c_ m$ is still present in the forward pass; it’s simply not amplified via CFG, explaining why motion trajectories are still observed.
>
> **(5) Wan 2.2 (5B) performing worse than Wan 2.1 (1.3B)**
>
> This is an interesting phenomenon we also observed. Most concurrent works and the community currently adopt Wan 2.1 (1.3B) for its balance of performance and efficiency. As discussed in the Supplementary A., Wan 2.2 (5B) is a very experimental model using a highly compressed 16x16x4 latent space to support 720p videos at 121 frames (vs 81 frames for all other Wan 2.1 and 2.2 variants) and is the only model in Wan family to support T2V and I2V (TI2V) within a single model. While this compact latent space enables 720p generation at 24 FPS, we hypothesize that reasoning on such compressed latents presents challenges for the 5B-scale model which has to handle longer sequences at higher resolution, supporting both T2V and I2V. We observe that the 5B model generates exceptional details with dense, accurate motion signals, yet generalizes worse than the 1.3B model on sparse tracks or inaccurate user drags. Architecturally, Wan 2.1 (1.3B) includes an image cross-attention module that helps preserve the original structure, which the unified Wan 2.2 lacks. Thus, we use the 1.3B model for most demos. The 5B model is recommended for high-resolution tasks with dense inputs. Please refer to the “Baseline Comparison” tab of the website and zoom in for 5B videos, which have exceptional details compared to other baselines. Notably, since benchmarks like DAVIS are typically 480p, the 5B model’s 720p capability adds limited value to metrics.
>
> [1] AIGC-Apps and Alibaba PAI Team, “VideoX-Fun (Wan Fun)”, Github Repository.
>
> [2] Choi et al., “Enhancing Motion Dynamics of Image-to-Video Models via Adaptive Low-Pass Guidance”, arXiv 2025.

---

### Official Review · Reviewer_g9qk · 2025-11-01

**Soundness:** 3
**Presentation:** 3
**Contribution:** 3
**Rating:** 6
**Confidence:** 4

**Summary:**

This paper introduces *MotionStream*, a real-time video generation framework with interactive motion control. The authors distill a bidirectional diffusion teacher into a causal autoregressive model, achieving streaming generation up to 24 FPS on a single GPU. The paper further proposes attention sinks and rolling KV caches to enable stable long-sequence generation without increasing latency. Experiments demonstrate competitive visual quality and strong speedups across several motion-controlled generation tasks.

**Strengths:**

1. Addresses a timely and important problem: bringing real-time interaction to diffusion-based video models.

2. The proposed attention sink and KV cache mechanisms are intuitively sound and practically effective.

3. Extensive experiments and ablations support the framework's stability and efficiency.

4. Overall writing and presentation are polished and easy to follow.

**Weaknesses:**

1. The evaluation datasets appear limited to short human-action or camera-move clips; it's unclear how the model performs on scenes with large physical transformations (e.g., object deformation, rapid perspective change).

2. Limited theoretical analysis; most findings are empirical.

**Questions:**

See  Weakness

---

> ### Author Response · Authors · 2025-11-22
>
> Dear reviewer g9qk, we appreciate your comments and the time spent to review our work. Please refer to the general response for a summary of contributions, the link to our new [website](https://anonysubmission7ei3vc8j3x1f3d-spec.github.io/anony-submission-suppleweb/) containing diverse sample videos, and the updated manuscript. Here, we address your specific concerns.
>
> **(1) Results on large physical deformations and rapid perspective changes**
>
> To provide a rigorous and fair evaluation, we utilized the DAVIS dataset, a standard benchmark for motion control [1, 2], and additionally adopted high-quality demo videos from SORA 1’s initial demo to ensure a comprehensive assessment. We gently note that our evaluation actually covers a wider range of samples than many prior motion-control works which focus on a single dataset.
>
> While not shown in the paper, both DAVIS and SORA datasets contain challenging scenarios with rapid motion, significant perspective changes, and large object deformations. Please refer to the “Additional Evaluation Examples” tab in the webpage, where we elaborate numerous cases. There are also several demo examples with challenging drags (please check “wave”, “hot air balloons" and “mammoth” samples).
>
> Generally, all motion control methods (including ours) are susceptible to noisy conditioning signals. When scenes change abruptly or tracks become undefined (or vanish completely), the base model must infer the missing information. For this reason, MotionStream was primarily designed to animate objects inside the initial image rather than tasks like world exploration (track isn’t a good representation in those tasks too, as they would quickly become invisible with view changes). Since our data annotator, CoTracker3, also struggles under severe conditions, perfect motion preservation is inherently challenging.
>
> Nonetheless, our quantitative results confirm that MotionStream remains highly effective in animating most challenging motions compared to existing methods. We believe that adopting a larger and stronger backbone (e.g., Wan 14B) would further reduce artifacts, given its superior performance for generating natural scenes.
>
>
> **(2) Limited theoretical analysis**
>
> Our findings are established through a systematic analysis of emergent phenomena (e..g attention behaviour, which is also actively being discussed in the LLM community [3, 4]) and logical methodological development, such as properly simulating extrapolation distribution during training. As is common in most vision and LLMs research, we think some contributions are best justified through careful architecture probing and thorough experimental validations. We would be happy to provide further logical elaboration or proofs on any specific components if the reviewer would be interested.
>
> [1] Geng et al., “Motion Prompting: Controlling Video Generation with Motion Trajectories”, CVPR 2025
>
> [2] Lin et al., “Ctrl-Adapter: An Efficient and Versatile Framework for Adapting Diverse Controls to Any Diffusion Model”, ICLR 2025
>
> [3] Barbero et al., “Why do LLMs attend to the first token?”, COLM 2025
>
> [4] Gu et al, “When Attention Sink Emerges in Language Models: An Empirical View”, ICLR 2025

---

### Official Review · Reviewer_EzdD · 2025-11-02

**Soundness:** 2
**Presentation:** 2
**Contribution:** 2
**Rating:** 4
**Confidence:** 4

**Summary:**

This paper proposes a method named Motionstream, which helps interactive video  generation with flexible motion control and efficient inference speed. Specifically, Motionstream trains a teacher video generation model with both text and motion control and Motionstream distills the teacher with Diffusion strategy to the student with auto-regressive strategy. The student with auto-regressive strategy can generate the interactive video at constant high speed.  To further bridging the domain gap from training on finite length and extrapolate, and preventing error accumulations, MotionStream incorporates self-rollout with attention sinks and KV cache rolling during training. Finally, MotionStream is evaluated from respective of the motion following, video quality and inference efficiency.

**Strengths:**

1. Complete "Teacher-Student" Framework: The entire pipeline, from designing an efficient motion-controlled teacher model to distilling a causal student model, is rigorously and systematically designed.
2. Lightweight Track Encoder: The use of sinusoidal positional encoding with a learnable track head, compared to the VAE-based RGB encoding method, achieves a 40× speedup in encoding while maintaining high track adherence accuracy, which is crucial for real-time systems.
3. Attention Sinking with Rolling KV Cache: This is one of the core innovations of the paper. The authors adapt the "attention sink" concept from LLMs and successfully apply it to video diffusion models. By using the initial frame as a fixed "anchor" combined with a local rolling context window, it effectively solves the drift problem in long-video generation, ensuring long-term stability. The paper explicitly highlights that its method eliminates the train-test discrepancy by simulating inference-time extrapolation during training, which offers an advantage over contemporaneous work.

**Weaknesses:**

1. It is worth noting that utilizing distillation to enhance model inference efficiency is an established technique in model optimization, including the video generation domain. Several contemporaneous works, such as Hunyuan-Gamecraft and notably Matrix-Game 2.0, have adopted a highly similar paradigm—combining a distillation framework with an autoregressive student model and Self-Forcing training. While the integration presented here is non-trivial, the core methodological concept bears a strong resemblance to these existing approaches, which consequently lowers the perceived novelty of the contribution
2. Furthermore, while the paper highlights motion control as a key contribution alongside the distillation framework, the synergistic relationship between the two remains unclear. They are presented as somewhat decoupled components. The authors should articulate a more cohesive narrative that explicitly links the design of the motion control mechanism to the requirements and success of the causal, distilled student model.

**Questions:**

I find the overall approach of MotionStream to be very similar to that of Matrix-Game 2.0, which leads me to question the novelty of this work's contribution.

---

> ### Author Response · Authors · 2025-11-22
>
> Dear reviewer EzdD, we greatly appreciate your perspective and efforts in reviewing our work. Please refer to the general response for summary of contributions, the new [website](https://anonysubmission7ei3vc8j3x1f3d-spec.github.io/anony-submission-suppleweb/) with diverse sample videos, and an updated manuscript. Here, we address your specific concerns.
>
> **(1) Contemporaneous works and the novelty of the distillation framework**
>
> We respectfully note that Matrix-Game 2.0 (Aug 12, 2025) and Hunyuan-Gamecraft (Aug 21, 2025) are both concurrent to ours and they have not been published in any academic conference or journal. We believe these similarities should not diminish the novelty of our contributions. Furthermore, as recognized by the reviewer, the difference is non-trivial and we have clearly distinct goals.
>
> * Both Matrix-Game 2.0 and Hunyuan-Gamecraft aim at developing action (keyboard) conditioned world models specialized in game-like domains (e.g., Minecraft) and are trained with larger data and compute. On the other hand, MotionStream focuses on animating open-domain real-world images using motion (track) conditions. Please refer to the website for MotionStream’s diverse use cases.
>
> * Compared to Matrix-Game 2.0, our training systematically addresses long-video drifting issues via attention sink and window size analysis. Combined with a small attention window and efficient VAE decoder our method allows generation of longer videos at much faster speeds (Ours: 832x480 @29.5 FPS, 1280x704 @24FPS, Matrix-Game: 640x352 @25FPS).
>
> Many of the previous/concurrent video/world model works build upon teacher training and (causal) distillation paradigm. However, we believe MotionStream has a different goal and design as summarized in the general comment (such as efficient motion condition or long video extrapolation) and strong practical applications can be built on top of it.
>
>
> **(2) Motion control and the distillation framework**
>
> We thank the reviewer for valuable feedback. We’d like to clarify how our design harmonizes motion control with distillation. We recognize that without architectural efficiency, few-step causal distillation remains bottlenecked by heavy conditioning modules (e.g., ControlNets that double FLOPs and slow video VAEs). Our lightweight trackhead design and channel-wise concatenation explicitly address this overhead. We then distill our optimized joint CFG, derived from analyzing the trade-off between motion adherence and quality, directly into the student via the DMD: $\nabla_ \theta \mathcal{L}_ {\text{DMD}} \approx -\mathbb{E}_ {t, \hat{z}_ 0} \left[ ( s_ {\text{real}} - s_ {\text{fake}} ) \cdot \frac{\partial \hat{z}_ 0}{\partial \theta} \right]$. Here, the teacher target $s_ {\text{real}}$ incorporates joint guidance via the frozen teacher $f_{\phi}$: $s_ {\text{real}} := s_ {\text{base}} + w_t \cdot (f_{\phi}(c_t, c_m) - f_{\phi}(\emptyset, c_m)) + w_m \cdot (f_{\phi}(c_t, c_m) - f_{\phi}(c_t, \emptyset))$, where $s_ {\text{base}} = \alpha \cdot f_{\phi}(\emptyset, c_m) + (1-\alpha) \cdot f_{\phi}(c_t, \emptyset)$ with $\alpha = w_t/(w_t + w_m)$ and “fake” score is approximated with a trainable critic $s_ {\text{fake}} := f_{\psi}(c_t, c_m)$ without CFG. By intentionally applying joint CFG only to the teacher target, our few-step student $G_ \theta$ receives informative gradients encapsulating benefits of both motion and text guidance within a single NFE. We have included these details in the manuscript as well. As such, we politely note that MotionStream has several elements that effectively blend motion conditions both architecturally, and during causal distillation. We have  improved the overall manuscript for clarity, but let us know if you would like to have specific sections edited.

---

### Author Response · Authors · 2025-11-22
**General response to all reviewers [1]**

We thank all reviewers for their valuable comments and constructive feedback. We are grateful that reviewers unanimously recognized our work with the following strengths: (1) well-motivated approach to an important problem, (2) technically sound pipeline design (including architecture, attention design, and causal distillation), and (3) polished presentation with thorough experiments.

As a general comment, we would like to clarify the core contributions of our work and provide additional qualitative examples through an anonymized [website](https://anonysubmission7ei3vc8j3x1f3d-spec.github.io/anony-submission-suppleweb/) (https://anonysubmission7ei3vc8j3x1f3d-spec.github.io/anony-submission-suppleweb/).

**(1) Revisiting the contribution of MotionStream**
* **First real-time motion-controllable video generation system.** To our knowledge, MotionStream is the first real-time motion (track) conditioned video generation pipeline capable of running on a single GPU. Unlike prior works, MotionStream allows users to “play with the video model”: users can paint trajectories, control cameras, or transfer motion and watch results streaming in real-time. As demonstrated in our additional results, the frontend UI (featuring diverse functions such as static grids and multiple moving points) shows high potential for interactive content creation.
* **Efficient pipeline design.**
While motion control and distillation may seem distinct, they are closely interleaved to achieve high throughput and low latency crucial for interactive response. In particular, our efficient track-condition head achieves 40x faster encoding speed than prior RGB VAE-based approaches [1, 2], and our channel-wise concatenation design halves FLOPs compared to prior ControlNet architectures [1, 3]. In addition, baking text-motion joint guidance into Self Forcing-style distillation’s target keeps student NFE to 1 while preserving strong motion and text following capability. Moreover, we propose to train an extra tiny VAE that runs significantly faster than the original Video VAE for the purpose of reducing decoding time. Consequently, our design enables 29.5 FPS at 480p and 24 FPS at 720p on a single H100 GPU, the fastest among all prior or even concurrent works (including T2V). Notably, our design minimizes the compute overhead of motion conditioning modules, which are commonly expensive in other methods, creating a synergistic design with proper extrapolation-aware causal distillation.
* **Long video generation.** We systematically analyze the effect of attention sinks and local attention windows, proposing a distillation method that explicitly simulates extrapolation distributions. To our best knowledge, it’s the first work in the video diffusion domain, to explore the impact of attention sinks with proper extrapolation-aware training. Compared to T2V models, which can easily perform long self-rollouts during training, it is harder to simulate very long videos with corresponding long-spanning motion tracks. Our approach enables training on short sequences while generalizing to arbitrarily long videos. This could be more beneficial for control-based autoregressive video generation systems where obtaining accurate long control signals is difficult. We showcase a sample in a demo where video quality is maintained even after 5,000 frames.
* **State-of-the-art quality & Generalizability.** Our method delivers state-of-the-art results in motion control and camera control benchmarks, and generalizes robustly to user interactions. Our entire training pipeline (from teacher training to full causal distillation) is fully based on publicly available models and datasets, and requires only 4 days on 32 A100 GPUs, which is highly efficient given the massive compute loads of recent large models.

**(2) Website with additional samples**
To provide diverse samples requested by reviewers, we opened a [website](https://anonysubmission7ei3vc8j3x1f3d-spec.github.io/anony-submission-suppleweb/) containing extensive use cases, including diverse demo samples, camera controls, and motion transfer. Many samples extend beyond 500 frames, and several contain rapid motions (e.g., the "wave" example where the mouse is dragged really quickly). Most user-defined trajectories are not pixel-wise accurate (flat grids with full visibility), yet the model generalizes effectively. We additionally provide challenging samples from our DAVIS and SORA evaluation dataset, I2V results (where the model receives no motion condition), and most importantly failure cases to provide a transparent view of our model’s limitations.

[1] Gu et al., “Diffusion as Shader: 3D-aware Video Diffusion for Versatile Video Generation Control”, SIGGRAPH 2025

[2] AIGC-Apps and Alibaba PAI Team, “VideoX-Fun (Wan Fun)”, Github Repository.

[3] Geng et al., “Motion Prompting: Controlling Video Generation with Motion Trajectories”, CVPR 2025

---

### Author Response · Authors · 2025-11-22
**General response to all reviewers [2]**

We gently note that our system achieves 29.5 FPS with 0.4s latency on a single H100 using Flash Attention 3 (metrics in our initial paper were reported with FA2), marking the fastest speed among previous/concurrent works. This responsiveness is critical for interactive motion control. We have updated our manuscript's speed metrics accordingly. Additionally, we have uploaded a revised manuscript incorporating (1) slightly polished writing combined with reviewers’ suggestions, and (2) additional details in the supplementary, including failure cases, backbone-specific behaviours, application in drag-based image editing, and demo details.

We appreciate all reviewers and AC’s efforts in shaping a better manuscript and look forward to further discussions.

---

### Author Response · Authors · 2025-12-03
**Summary of Rebuttal**

Dear Area Chair,

We understand the unexpected challenges caused by the recent OpenReview leak and sincerely appreciate your extra efforts. With our initial rebuttal posted on Nov 21, we were planning to leave a follow-up reminder a week after (Nov 28), but the freeze unfortunately halted the discussion phase. It is regrettable that this interruption occurred during the final days which are pivotal for reviewer engagement, precluding the opportunity to solidify a stronger consensus.

Nonetheless, we believe our reviewers recognized the strengths of our work, and our response effectively addressed the reviewer’s initial reservations. We provide a summary of rebuttal discussions below to assist your assessment. Our supplementary [website](https://anonysubmission7ei3vc8j3x1f3d-spec.github.io/anony-submission-suppleweb/) provides real-world use cases and additional examples requested by the reviewers, and the manuscript has been revised accordingly.

**Reviewer EzdD:** (1) Clarified distinctions between our work and concurrent approaches regarding scope and design goals; (2) Articulated more synergistic integration of architectural efficiency and motion control within the causal distillation framework and improved the narrative for clarity in the revision.

**Reviewer g9qk:** (1) Demonstrated robustness in challenging scenarios (e.g., rapid motion, perspective changes) via extensive results on the website and added discussion; (2) Clarified how our architecture design and causal distillation were built on systematic analysis and logical foundations.

**Reviewer 5Qbe:** (1) Revised manuscript for better elaboration of the contributions; (2) Validated that our conditioning strategy preserves the backbone model’s generative capability and I2V performance via new quantitative experiments; (3) Provided more results on several corner cases as requested; (4) Explained why text-only CFG can still lead to motion following; (5) Added discussions on Wan 2.1 (1.3B) and Wan 2.2 (5B)’s architectural differences and our observations on their performance.

**Reviewer RuEW:** (1) Better articulated our contributions and technical novelty; (2) Provided long-form samples (up to 5000 frames) and scene-change examples to illustrate the effect of attention sink; (3) Justified the different CFG scales for text and motion through grid search provided in the paper.

We are confident that MotionStream represents a significant step forward in transforming video generation models into a truly interactive interface. We hope the AC will consider the strength of our rebuttal and the value demonstrated in our submission. Please check the corresponding threads for the detailed responses. Thank you for your time and dedication to the community.



Best regards,

The Authors

---

### Meta-Review · Area_Chair_qBe4 · 2025-12-10

**Summary:**

This paper presents MotionStream, a real-time, motion-controllable video generation system that achieves significant speed improvements while maintaining competitive quality and enabling interactive applications. The core contributions lie in the synergistic integration of an efficient motion-conditioning pipeline, causal distillation with joint guidance, and a novel training strategy employing attention sinks with rolling KV caches to enable stable, long-horizon generation.

Four reviewers provided detailed assessments, with initial ratings spanning from marginally below to marginally above the acceptance threshold. Common concerns:

- Novelty: Reviewers EzdD and 5Qbe felt the method appeared as a composition of known techniques (distillation, attention sinks, motion conditioning), questioning the distinctness from concurrent works like Matrix-Game 2.0.

- Evaluation: Reviewers g9qk, 5Qbe, and RuEW requested more evidence on performance under challenging conditions (large deformations, rapid motions, long sequences) and a clearer analysis of trade-offs (e.g., impact on base model's generative capability).

- Technical: Some reviewers noted a reliance on empirical results over theoretical analysis and sought clarification on specific design choices (e.g., guidance scaling, model size performance).

**Reviewer Concerns:**

The author's response was highly effective in mitigating the major concerns:

- Novelty and contribution are now significantly clearer and well-articulated. The work's unique positioning—bridging efficient motion control, real-time distillation, and long-horizon stability for interactive open-domain video animation—is convincingly presented. The distinctions from concurrent works are valid and important for the community.

- Empirical evaluation is substantially strengthened by the new website and quantitative I2V analysis. While not all evaluation desires (e.g., theoretical analysis) are fully met, the provided evidence robustly supports the paper's claims regarding performance, generalization, and limitations.

- Technical soundness remains a strength, and the rebuttal clarified key design rationales. The core methodological integration is non-trivial and effectively solves a practical problem.

**Reviewer Scores:**

##Reviewer EzdD
Initial Score: 4 (marginally below acceptance threshold)

Confidence: 4 (confident but not absolutely certain)

Likely Score Change: 6

Reasoning:
This reviewer’s main concern was novelty, specifically the similarity to Matrix‑Game 2.0. The authors convincingly differentiated their work by emphasizing: (1) open‑domain real‑world motion animation vs. game‑specific world models, (2) systematic attention‑sink analysis for long‑video stability, (3) faster speeds and higher resolution. The additional website with diverse samples and the I2V ablation also addressed the “synergy” concern. While the reviewer might still view the core distillation framework as not radically novel, the unique combination for real‑time interactive video control likely meets the bar for acceptance. A move to a weak accept (5 or 6) is plausible.

##Reviewer g9qk
Initial Score: 6 (marginally above acceptance threshold, but wouldn’t mind rejection)

Confidence: 4 (confident but not absolutely certain)

Likely Score Change: 6 or 8

Reasoning:
This reviewer’s concerns were about evaluation on challenging motions and lack of theoretical analysis. The authors provided extensive new qualitative evidence (website with rapid motions, deformations, long sequences) and justified the empirical approach as common in vision/LLM research. Given the reviewer’s positive tone (“timely and important problem,” “sound and effective” methods) and the direct response to their weakness, they would likely become more confident in the paper’s contribution, raising the score to a clear accept.

##Reviewer 5Qbe
Initial Score: 6 (marginally above acceptance threshold, but wouldn’t mind rejection)

Confidence: 3 (fairly confident)

Likely Score Change: 6 or 8

Reasoning:
The reviewer had several specific concerns: contribution clarity, trade‑offs of motion conditioning, missing I2V comparison, and evaluation on corner cases. The authors addressed each directly: Clarified contributions in the general response. Added a quantitative I2V table showing no degradation from motion injection. Provided website with challenging and failure cases. Explained text‑only guidance and model‑size results. This thorough rebuttal likely resolves the major hesitations, moving the reviewer to a solid accept.

##Reviewer RuEW
Initial Score: 6 (marginally above acceptance threshold, but wouldn’t mind rejection)

Confidence: 3 (fairly confident)

Likely Score Change: 6 or 8

Reasoning:
Concerns were about the technical contribution being a “straightforward combination” and limited long‑form examples. The authors emphasized the non‑trivial integration for real‑time performance and provided many long‑form demos (300‑500+ frames) plus failure cases. The reviewer might still consider the core ideas incremental, but the demonstrated practical impact (real‑time interaction, infinite‑horizon generation) and comprehensive evaluation could lift the score to a more confident weak accept or clear accept.

---

### Decision · Program_Chairs · 2026-01-26

Accept (Oral)